# Transfer learning guided discovery of efficient perovskite oxide for alkaline water oxidation

Chang Jiang[1,8], Hongyuan He[2,8], Hongquan Guo[1], Xiaoxin Zhang[1], Qingyang Han[1], Yanhong Weng[3], Xianzhu Fu [3], Yinlong Zhu [4], Ning Yan [5], Xin Tu [2] ✉ & Yifei Sun [1,6,7] ✉

Perovskite oxides show promise for the oxygen evolution reaction. However, numerical chemical compositions remain unexplored due to inefficient trial-and-error methods for material discovery. Here, we develop a transfer learning paradigm incorporating a pre-trained model, ensemble learning, and active learning, enabling the prediction of undiscovered perovskite oxides with enhanced generalizability for this reaction. Screening 16,050 compositions leads to the identification and synthesis of 36 new perovskite oxides, including 13 pure perovskite structures. $Pr_{0.1}Sr_{0.9}Co_{0.5}Fe_{0.5}O_3$ and $Pr_{0.1}Sr_{0.9}Co_{0.5}Fe_{0.3}Mn_{0.2}O_3$ exhibit low overpotentials of 327 mV and 315 mV at 10 mA cm$^{-2}$, respectively. Electrochemical measurements reveal coexistence of absorbate evolution and lattice oxygen mechanisms for O-O coupling in both materials. $Pr_{0.1}Sr_{0.9}Co_{0.5}Fe_{0.3}Mn_{0.2}O_3$ demonstrates enhanced OH$^-$ affinity compared to $Pr_{0.1}Sr_{0.9}Co_{0.5}Fe_{0.5}O_3$, with the emergence of oxo-bridged Mn-Co conjugate facilitating charge redistribution and dynamic reversibility of $O_{lattice}/V_O$, thereby slowing down Co dissolution. This work paves the way for accelerated discovery and development of high-performance perovskite oxide electrocatalysts for this reaction.

Perovskite oxide materials play a pivotal role in the green electrosynthesis of value-added chemicals, a critical step towards achieving carbon-neutrality[1,2]. Their remarkable compositional tunability at the atomic scale and low cost for large-scale production have propelled them to the forefront of research. One particularly noteworthy application is their utilization in the oxygen evolution reaction (OER), which can be coupled with various cathodic reactions such as the hydrogen evolution reaction[3], CO$_2$ reduction reaction (CO$_2$RR)[4], and N$_2$ reduction reaction (N$_2$RR)[5]. However, OER remains kinetically sluggish, involving a 4-step proton-electron coupled transfer process.

Therefore, the development of highly efficient and cost-effective electrocatalysts is of paramount importance[6,7].

Previous studies have established that incorporating various cations (Ce, Pr, Cr, Sr, V, W, Co, Fe, Mn, Nb, Mg etc.) into either the A-site or B-site of perovskite oxide can effectively modulate the local coordination environment and electronic structure, leading to enhanced electrocatalytic performances[8]. However, the large number of potential compositions that arise from this combinatorial approach necessitates a paradigm shift away from traditional research strategies that rely solely on trial-and-error experiments or unpredictable

[1]College of Energy, Xiamen University, Xiamen, China. [2]Department of Electrical Engineering and Electronics, University of Liverpool, Liverpool, UK. [3]Shenzhen Key Laboratory of Energy Electrocatalytic Materials, Guangdong Research Center for Interfacial Engineering of Functional Materials, College of Materials Science and Engineering, Shenzhen University, Shenzhen, China. [4]Institute for Frontier Science, Nanjing University of Aeronautics and Astronautics, Nanjing, China. [5]School of Physics and Technology, Wuhan University, Wuhan, China. [6]State Key Laboratory of Physical Chemistry of Solid Surface, Xiamen University, Xiamen, China. [7]Shenzhen Research, Institute of Xiamen University, Shenzhen, China. [8]These authors contributed equally: Chang Jiang, Hongyuan He. ✉e-mail: xin.tu@liv.ac.uk; yfsun@xmu.edu.cn

serendipitous discoveries. To accelerate the exploration of novel and efficient perovskite oxide-based OER catalysts, high-throughput density functional theory (DFT) calculations have emerged as a promising alternative. Nevertheless, these calculations often require prior knowledge of specific algorithms or methods, hindering data unification across different systems and limiting the universal applicability of the results under various conditions[9]. Additionally, while computational costs are lower than experimental work, they are not negligible, further complicating seamless plug-and-play implementation.

Recent advancements in artificial intelligence (AI) have unveiled its remarkable potential in the discovery of novel electrocatalysts[10]. Machine learning (ML) offers a data-driven methodology for extracting quantitative composition-catalytic property relationships, holding the promise to expedite materials design by orders of magnitude[11,12]. The majority of data inputs for ML models typically stem from published experimental data/open-source databases, serving as a cornerstone for the generation of derivative new candidates[13,14]. Some efforts have focused on using ML to identify highly relevant descriptors, thereby simplifying the system and accelerating the prediction process[15–17]. However, the ML algorithms based on feature selection and simplification often eliminate less-significant descriptors, inevitably leading to information loss and diminished prediction accuracy. Moreover, when analyzing the relative importance of the same dataset, different algorithms frequently produce inconsistent results[18]. Consequently, the reliability of feature importance interpretations in the absence of domain expertise is often questionable[19,20].

In addition to the choice of algorithms, the quality and quantity of data play a crucial role in determining the accuracy of ML-based predictions. Conventional simulation databases derived from DFT are often limited to a single or a few similar systems, restricting the range of applicability and generalizability of the extracted knowledge. Moreover, experimental data, while valuable, are relatively scarce and challenging to consolidate due to the lack of universally accepted standards for reporting experimental methodologies. Despite recent advancements in unifying inconsistently reported data from diverse sources, these efforts remain limited to specific reactions and often involve feature selection, resulting in trimmed feature sets that may exclude potentially valuable information[21].

To tackle this challenge, herein, we propose a powerful transfer learning paradigm centered on the cation information of perovskite oxide electrocatalysts. This approach leverages pre-trained models to effectively integrate OER data with vast datasets from diverse research fields, encompassing a broader range of perovskite compositions. Additionally, we employ ensemble methodologies to combine models derived from distinct sub-clusters identified through a combination of domain knowledge and unsupervised learning techniques. This strategy facilitates the transfer of knowledge across diverse material systems, leading to a significant improvement in prediction accuracy. Furthermore, the incorporation of active learning methodology expands the predictive capabilities to more complex hexanary material systems, extending beyond the initial focus on quinary systems. The $Pr_{0.1}Sr_{0.9}Co_{0.5}Fe_{0.5}O_3$ (PSCF) and $Pr_{0.1}Sr_{0.9}Co_{0.5}Fe_{0.3}Mn_{0.2}O_3$ (PSCFM) demonstrated remarkable OER activity, with low over-potentials below 330 mV (at 10 mA cm$^{-2}$ in 1 M KOH). Comprehensive characterizations reveal that lattice oxygen plays a key role in facilitating O-O coupling during the OER process. DFT calculations further elucidate the mechanistic underpinnings of this enhanced OER activity. The incorporation of Mn into PSCF strengthens the stability of Co reactive sites, while simultaneously lowering the reaction barrier through the lattice oxygen mechanism (LOM) pathway on Mn-O-Co motifs. Our approach demonstrates the effectiveness of transfer learning and active learning in overcoming data limitations and achieving accurate prediction of OER catalysts.

## Results

### Transfer learning methodology for prediction

**Data extraction and cation encoding.** The proposed transfer learning loop, as depicted in Fig. 1, comprises seven steps: data extraction, cation encoding, feature embedding, clustering, local prediction, global ensembling, and experimental validation with active learning closed loop. Due to the limited availability of perovskite oxide data for OER, we additionally collected data for non-OER perovskite oxides (see Supplementary Data 1 for a comprehensive list of data collected and predicted in this work). This approach expanded the dataset by 48.9% from 94 to 140 entries (Fig. 2a). The enriched dataset encompasses a diverse range of features, including material compositions, oxygen vacancy concentration, and chemical valence state distributions.

### Feature embedding

We employed a pre-training process that utilizes an auto-encoder with shortcut connections (AESC)[22] to project cation encoding into embeddings with reduced dimensionality while preserving informational integrity, thereby enhancing the data density. We used an n-fold cross-validation methodology to identify the most effective AESC architecture, evaluating a range of configurations from simplistic to intricate. Subsequently, we synthesized these sub-models using an ensemble strategy by computing the arithmetic mean of all predictions. The performance evaluation matrix (Supplementary Note 2) employed the correlation of determination ($R^2$) to access the correlation between the encoder inputs and the decoder outputs (reconstruction). A two-dimensional embedding was selected to maintain a balance between model performance and visualization requirements. Screened from 400 different models, the ensemble model constructed through 10-fold cross-validation, demonstrated an impressive $R^2$ of 0.77 and 0.96 on the test and the entire dataset, respectively (See Fig. 2c and Supplementary Fig. 1 for the full results). Furthermore, to evaluate the efficacy of transfer learning, the ability of an AESC to capture different classes of material information was measured using the reconstruction root mean square error (RMSE). As illustrated in Fig. 2d, the proposed AESC demonstrates superior reconstruction fidelity on non-OER datasets, achieving a significant reduction in RMSE from 3.25 to 1.05. This compelling result suggests that material embeddings extracted through transfer learning approaches are effective in capturing the characteristics of unreported OER candidates. Notwithstanding, the findings in Fig. 2d alone are insufficient to conclusively establish the efficacy of transfer learning; its robustness must be further rigorously evaluated through comprehensive experimentation.

### Clustering and location prediction

We subsequently employed the embeddings of OER materials, along with their corresponding reaction conditions, as input variables for the model. 91 out of a total of 94 points were included, and the rest 3 points were excluded due to their uncertainty in the reaction conditions. The model was then trained to predict the overpotential at a current density of 10 mA cm$^{-2}$. A novel ensemble model of Gradient Boosting Regressor (GBR)[23] was implemented, with hyperparameter optimization conducted through a grid search of a consistent set of hyperparameters (48,600 different combinations for each model, see Supplementary Note 3 for the full list). However, visualization revealed an uneven distribution of data in the 2D embedding space. A model indiscriminately trained on the entire dataset exhibited low accuracy, with an RMSE of 42.79 mV, highlighting the inconsistency in OER performance attributable to the inherent complexity of our material system. To address this issue, we employed a data segmentation strategy. The data was divided into clusters characterized by greater internal consistency, allowing for individualized model training for each cluster. An unsupervised learning method, K-Means clustering in Lloyd style[24] was utilized to classify the data into multiple clusters

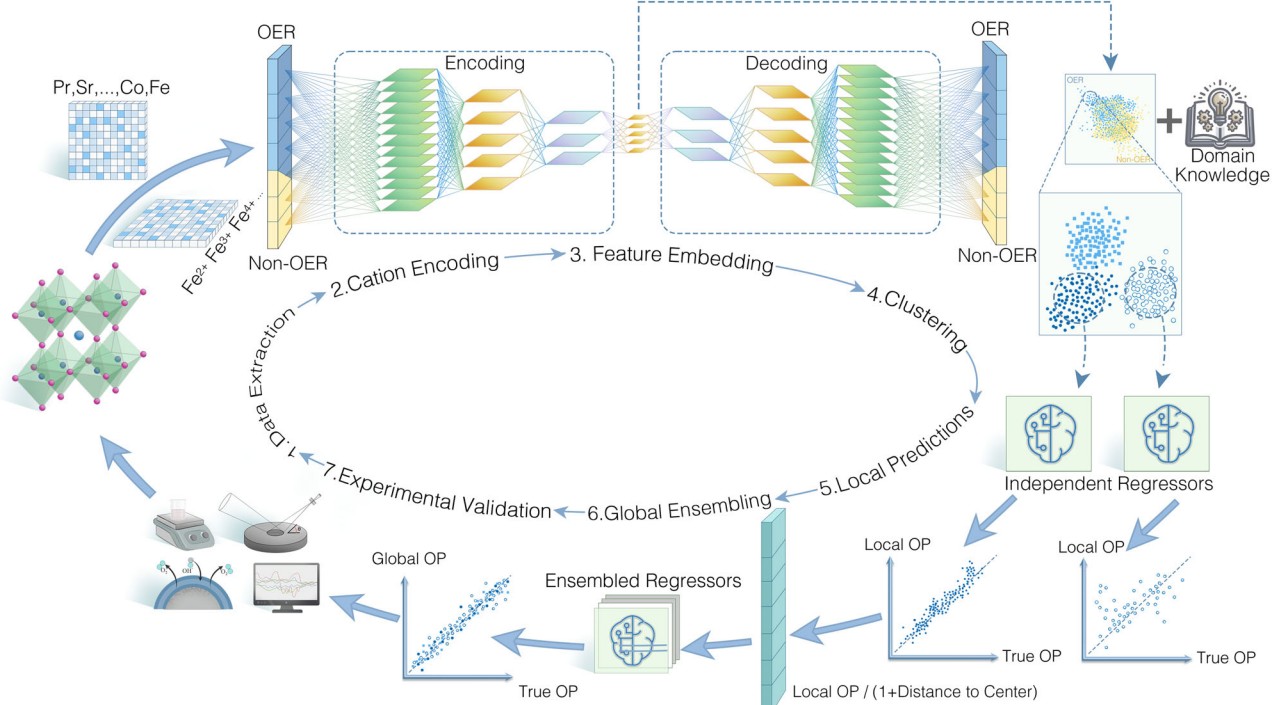

**Fig. 1 | Transfer learning workflow to discover perovskite electrocatalysts for the oxygen evolution reaction.** The OER-specific data subset is more extensive than the non-OER subset, incorporating various parameters for the characterization of the OER catalysts, including mass loading (in mg cm$^{-2}$), electrolyte concentration (in mol L$^{-1}$), substrate type (rotating disk electrode, glassy carbon or glassy carbon electrode), and the overpotential required to achieve a current density of 10 mA cm$^{-2}$ (in mV). All catalysts were synthesized using the sol-gel method. While previous studies reported negligible variations in individual features when employing chemical valence state distributions for feature encoding[62,63], our findings indicate that a specific material composition can be linked to a broad range of cation-encoded data points. For example, a perovskite composition (Co/Fe in B-site) with 10 distinct chemical valence distributions, could generate 419,281 data points (Fig. 2b). Such substantial cumulative discrepancies would significantly reduce the accuracy of the data. Therefore, we computed 20 intrinsic features for each perovskite composition based on cation encoding (see Supplementary Note 1 for computational details).

based on Euclidean distances (See Supplementary Figs. 2–4 for data visualization). This approach promotes homogeneity within clusters and facilitates effective GBR training for each cluster. Additionally, as an alternative to spatial-location based clustering, a sorting principle informed by domain knowledge (phase-sorting) was also considered. This method divided the data into seven categories: unknown, cubic, hexagonal, monoclinic, orthorhombic, rhombohedral, and tetragonal. To determine the optimal cluster number for K-Means clustering, we evaluated a range of cluster sizes from 1 to 19. Six different metrics were employed to guide this selection process (Fig. 2e). Examining the sum of squared distance of each data point to its nearest cluster center revealed a leveling off in the data once the cluster count reached 5 or 6, suggesting potential options for the number of clusters[25]. Additionally, we adopted phase-sorting as a benchmark for K-Means clustering. Based on a composite of metrics from various measurements, including Adjusted Rand Index[26], V-Measure, Completeness, Homogeneity[27] and Silhouette[28] scores (see Supplementary Note 4 for computational details), we selected cluster sizes of 5, 6, 13, 15, and 18 as they represent inflection points. Notably, none of these configurations achieved particularly high scores across all evaluation metrics. This unsatisfactory result may be attributed to the fact that a significant portion of the dataset (24 out of 94 entries) falls into the 'unknown' category in phase-sorting clustering. Therefore, relying solely on clustering analyses to determine the most effective clustering methodology is insufficient. This highlights the importance to employ predictive models trained on a variety of clustering approaches to achieve a more robust assessment.

We employed a 3-fold cross-validation approach to tune the hyperparameters of the GBR models for each cluster to predict OER activities. Subsequently, the optimized models were refitted to the corresponding dataset within each cluster without concerns about overfitting, based on the assumption of high intra-cluster consistency. For cluster comprising fewer than 3 data points, the arithmetic mean of the overpotential values was used as the predicted outcome. We opted for a 3-fold cross-validation strategy primarily because a higher number of folds could preclude smaller clusters from undergoing effective cross-validation. In instances where clusters comprise fewer than three data points, the arithmetic mean of the overpotentials inherent to the cluster was employed as a fixed output. Each model trained in this fashion was predicated on data exclusive to individual, independent clusters, a methodology we refer to as 'local prediction'. The most effective model was constructed based on 13 clusters for K-Means, achieving a relatively lower overall RMSE of 29.8 mV, while the model based on phase-sorting yields a RMSE of 33.45 mV (Fig. 2f). This slight improvement primarily arises from an inability of a singular clustering methodology to categorize all data points comprehensively, coupled with the fact that local prediction models do not accommodate intersecting clustering schemes.

## Global ensembling
In the realm of local prediction, individual cluster-based models operate autonomously, devoid of inter-cluster information exchange, leading to the underutilization of substantial datasets. To address this inefficiency, transfer learning is used to synergize models trained across disparate clusters. Rinehart et al. proposed a methodology capable of evaluating the effectiveness of transfer learning by quantifying the degree of similarity among datasets, a measure dependent on the discernment of domain-specific knowledge[29]. Rao et al. estimated prediction uncertainty through similarity, measured by the distance between estimated-training data points[30], proposing that prediction

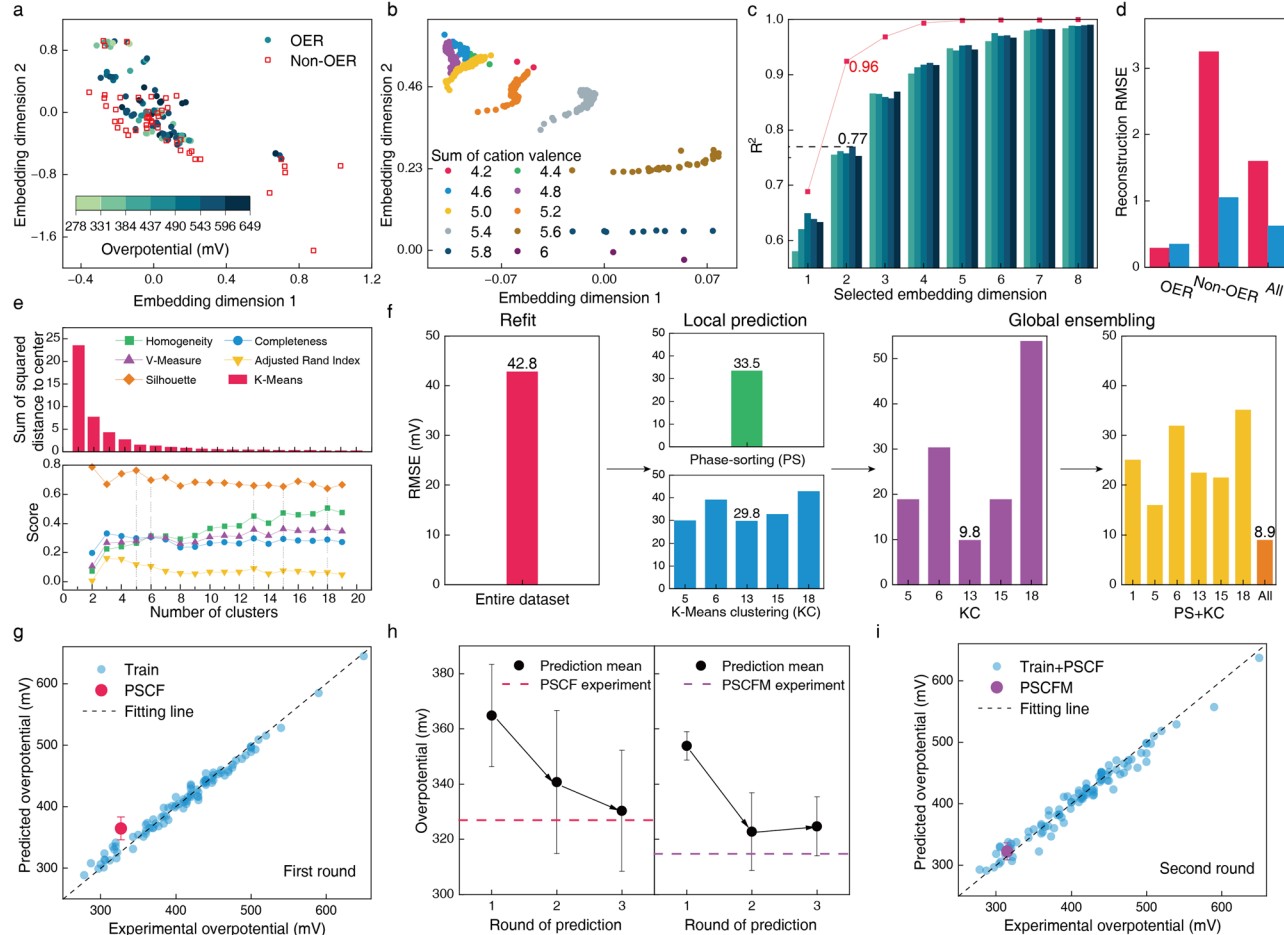

**Fig. 2 | Evaluation and prediction of transfer learning models. a** Distribution of OER data, associated OER activities (overpotential), juxtaposed with non-OER data across 2-D embeddings. **b** Visualization of cation encoding for a specific material formula, encompassing multiple hypothesized valence states, represented by 419,281 data points (sampling ratio of 1:1000 in the figure). **c** Comparative analysis of various embedding parameter settings. The gradation of bar colors from lighter to darker corresponds to the test set outcomes for 3, 5, 7, 10, and 15 folds cross-validation, respectively. The line graph delineates the ensemble outcomes of a 10-fold cross-validation conducted on the entire dataset. **d** The reconstruction RMSE

of the AESC. Red and blue bars denote AESC trained on OER data only and on both OER and non-OER data, respectively. The x-axis catalogs the datasets employed for computing the reconstruction RMSE. **e** Evaluation of K-Means clustering performance. **f** A comparative assessment of the prediction accuracy for OER activities. **g** First round of OER activities prediction. The error bar represents the standard deviation of the prediction. **h** Enhanced prediction accuracy attained through active learning processes for both PSCF and PSCFM. The error bar represents the standard deviation of the prediction. **i** Second-round of OER activities prediction. The error bar represents the standard deviation of the prediction.

accuracy diminishes as the distance between the prediction point and cluster center increases. In contrast to these approaches, we developed a global ensemble method that incorporates both domain knowledge and data distance metrics to assess data similarity. The similarity metric involves calculating the Euclidean distance between a predicted point and each cluster's centroid. Importantly, the relationship between similarity and prediction accuracy is not predetermined; instead, it is elucidated through the inductive capabilities of a ML model, termed global ensembling (Supplementary Note 5).

Global ensembling facilitates the transfer learning of data across various clusters, enabling a broader integration of cluster-specific information. Notably, when employing 13 clusters as the optimal configuration for local prediction, the RMSE substantially decreases from 29.8 mV to a global value of 9.80 mV (Fig. 2f). Furthermore, unlike in local prediction models where clusters operate independently, the global ensemble approach enables the integration of diverse clustering methodologies, as information could be transferred between spatial information and domain knowledge. This comprehensive incorporation of clustering data considerably enriches the informational substrate available for model learning, thereby enhancing the model's capacity for accurate fitting. The model combining phase-

sorting cluster, no clustering (1 cluster), K-Means 5, 6, 13, 15, and 18 clusters outperform the remaining local and global models with an RMSE of 8.90 mV and is selected as the final model to initialize the first-round prediction of novel materials. It is important to note that there are nuanced distinctions in the computation of the RMSE among direct refit, local prediction, and global ensemble strategies. The ultimate efficacy of the predictive model requires experimental validation.

## Experimental verification and active learning
We used this well-calibrated global ensemble model to conduct experimental validation of material candidates. Due to the inherent complexities associated with predicting the properties of perovskite oxide materials with higher structural entropy, our initial prediction was restricted to quaternary and quinary compositions (specifically, the overall cation type is 3 or 4: AA'BO or AA'BB'O, with the A site fixed to 6 different combinations, including Ba + Sr, La + Ca, La + Sr, Pr + Ba, Pr + Ca and Pr + Sr. The B site was selected from Co, Fe, Mn, Nb, Zr, Ni, Cu, Sn, Ir and Ru. A total of 7050 different formulas were generated in a combinatorial manner). Any particular A/B stoichiometry could correspond to multiple valence states, resulting in a range of predicted overpotential values. (See Supplementary Note 6 for speculative rules

concerning valence state distribution). Thirty chemical formulas were selected from over 5 million prediction points for experimental validation (Supplementary Fig. 5). Importantly, the material with the composition PSCF was predicted to be a high-performance material (Fig. 2g) with the lowest overpotential of 340.81 mV ($364.80 \pm 18.55$ mV). All samples were synthesized using the sol-gel method followed by annealing at 850 °C in air. Our X-ray diffraction (XRD) results confirmed the formation of 10 phase-pure perovskites (Supplementary Fig. 6a), while the remaining samples exhibited perovskite structures with significant impurities or intermediate phases (Supplementary Fig. 6b, c). Preliminarily the linear sweep voltammetry (LSV) evaluation confirmed an overpotential of 327 mV for PSCF. Subsequent X-ray photoelectron spectroscopy (XPS) characterization revealed the oxidative valence states of each cation in PSCF (detailed results are presented later in the paper). Thereafter, the data for PSCF were processed through cation encoding, feature embedding, and re-incorporated into our training sets for both local and global regressors in the second-round training iteration (Fig. 2h). This represents a typical active learning practice that involves augmenting the most recent experimental findings while continuously refining the model[30]. See Supplementary Figs. 7 and 8 for the second-round training results.

In the second round, a total of 9000 hexanary (AA'BB'B"O) formulas were generated, with the A sites being fixed with Pr + Sr, and the B sites incorporating three elements selected from the same list in the first-round prediction. This resulted in the shortlisting of 6 hexahydroxy formulas from an extensive dataset of over 20 million predicted data points (Supplementary Fig. 9). The PSCFM with Mn partially substituting Fe in PSCF was optimized from the second-cycle prediction, achieving a minimum predicted overpotential of 302.92 mV (322.75 mV $\pm$ 14.09 mV) (Fig. 2i). Subsequently, all these selected materials were fabricated, screened by XRD and evaluated by LSV measurement (Supplementary Fig. 6d). Consistent with the predictions, the PSCFM demonstrated a reduced overpotential of 315 mV at 10 mA cm$^{-2}$, validating the reliability of our model. Notably, without the implementation of active learning methodologies, the initially predicted mean performance of PSCFM in the first round was 353.82 mV (Fig. 2h), highlighting the crucial role of active learning in identifying such promising materials. Further validation of our active learning strategy involved incorporating precisely encoded valence state distributions of PSCFM into the training set for a third predictive cycle. See Supplementary Figs. 10 and 11 for the third-round training results.

The mean predictive error (MPE), defined as the discrepancy between the predicted average and the experimental values, demonstrated significant improvements across all three rounds of analysis for PSCF. MPE for PSCF decreased from an initial value of 37.91 mV in the first-round to 13.82 mV in the second round, ultimately reaching an impressively low 3.40 mV by the third round (Fig. 2h and Supplementary Fig. 12). For the PSCFM, the initial MPE of 39.07 mV was reduced to 9.91 mV in the third round, accompanied by a concurrent decrease in standard deviation by 3.38 mV (Fig. 2h). These findings suggest that despite the inherent complexity of hexanary systems, the application of active learning strategies facilitates a degree of convergence in predictive values, leading to enhanced predictive accuracy.

## Characterizations of predicted materials

The quantitively Rietveld refinement of the XRD patterns demonstrates that both PSCF and PSCFM primarily crystallize in a cubic phase with a Pm-3m space group (Fig. 3a). The scanning electron microscopy (SEM) images reveal the similar macroscopical morphology between PSCF and PSCFM (Supplementary Fig. 13). The transmission electron microscopy (TEM) image confirms the average particle size of ~100 nm. The atomic-scale high-angle annular dark field (HAADF)-scanning transmission electron microscopy (STEM) and elemental maps (Fig. 3b) confirm the uniform dispersion of all elements without observable phase segregation. The Pr and Mn atoms are located at the

positions of Sr and Co/Fe, respectively, indicating that Pr and Mn occupy the A and B sites of PSCFM.

As shown in Fig. 3c, the LSV curves clearly demonstrate that PSCFM and PSCF requires overpotential of only 315 mV and 327 mV, respectively, to achieve a current density of 10 mA cm$^{-2}$, outperforming commercial IrO$_2$ (375 mV). To compare the intrinsic activity of various electrocatalysts, the electrochemical double-layer capacitance (C$_{dl}$) and electrochemical active surface area (ECSA) were both measured and calculated (Fig. 3d, Supplementary Fig. 14 and Supplementary Table 7). The specific activity (defined as current density normalized on ECSA) of all catalysts is then calculated (Supplementary Fig. 15), which follows the order of PSCFM > PSCF > IrO$_2$ at a representative potential of 1.6 V. However, the ECSA values of all electrocatalysts are similar, suggesting that the significant difference in OER performance arises from the distinct reactivity of individual catalytic site. Consistent with this observation, the turnover frequency (TOF) curves also demonstrate the superior activity of PSCFM over PSCF and IrO$_2$ (Supplementary Fig. 16). Figure 3e demonstrates that PSCFM exhibits the smallest Tafel slope of 54.5 mV dec$^{-1}$, followed by PSCF (57.1 mV dec$^{-1}$), and commercial IrO$_2$ (75.4 mV dec$^{-1}$), indicating that the predicted electrocatalysts achieve favorable kinetics and accelerated electron transfer for OER.

Furthermore, in situ electrochemical impedance spectroscopy (EIS) was conducted at different potentials to elucidate the potential dependent charge-transfer kinetics during the OER (Fig. 3f, Supplementary Fig. 17 and Supplementary Table 8). The equivalent electric circuit consisted of electrolyte resistance (R$_s$) in series with two electrochemical resistors R$_{ct}$ (larger diameter of the low-frequency region) and R$_1$ (smaller diameter of the high-frequency region). The adsorption behavior of the reactants (OH$^-$) on the catalyst surface was described by R$_{ct}$ and CPE$_{ct}$. Evidently, the R$_{ct}$ with the adsorbed OH$^-$ intermediate species during OER predominately governs the variations in total charge transfer resistance. The PSCFM with a lower R$_{ct}$ at all potentials exhibits faster kinetics in the adsorption of OH$^-$ during OER. This is further corroborated by the Bode plot (Supplementary Fig. 18). The PSCFM delivers the highest frequency of 9.8 Hz (compared to IrO$_2$ at 4.8 Hz and PSCF at 8.6 Hz), confirming its fastest kinetics. Finally, the durability of the samples was evaluated by conducting galvanostatic tests in both three and two electrode modes.

The results reveal remarkable stability of PSCFM in both three-electrode and two-electrode electrolyzer configurations. In a three-electrode setup, the overpotential of PSCF and PSCFM exhibits only a ~3 mV fluctuation at a current density of 20 mA cm$^{-2}$ over a 20 h test (Fig. 3g). Furthermore, a two-electrode electrolyzer employing 20 wt% Pt/C supported carbon cloth as the cathode and PSCFM supported carbon cloth as the anode demonstrates stable electrolysis at 10 mA cm$^{-2}$ for ~50 h (Fig. 3h). Notably, the voltage increases with PSCFM (1.3%) is significantly lower compared to that with PSCF (2.3%), highlighting the superior durability of PSCFM in practical applications. The Faraday efficiency of PSCFM and PSCF anode catalysts was determined by measuring the concentration of gaseous products using gas chromatography based on an efficient four-electron reaction process. As shown in Supplementary Fig. 19 and Supplementary Table 10, the high Faraday efficiency over 96% could be achieved for all measured points, suggesting the good OER selectivity of both electrocatalysts. The durability of PSCFM was also evaluated in an alkaline water electrolyzer (membrane electrode assembly (MEA) mode) (Fig. 3i, j). Clearly, the electrolyzer equipped with a PSCFM anode and a Pt/C cathode exhibited only slight voltage vibration during continuous galvanostatic measurement at 30 mA cm$^{-2}$ and 50 mA cm$^{-2}$ for at least 80 h, implying its promising stability for practical applications.

## Dissection of structure and electronic configuration

To elucidate the intrinsic factors underlying their exceptional performance, the electronic configurations of both electrocatalysts were analyzed using XPS and X-ray absorption spectroscopy (XAS). The XPS

 

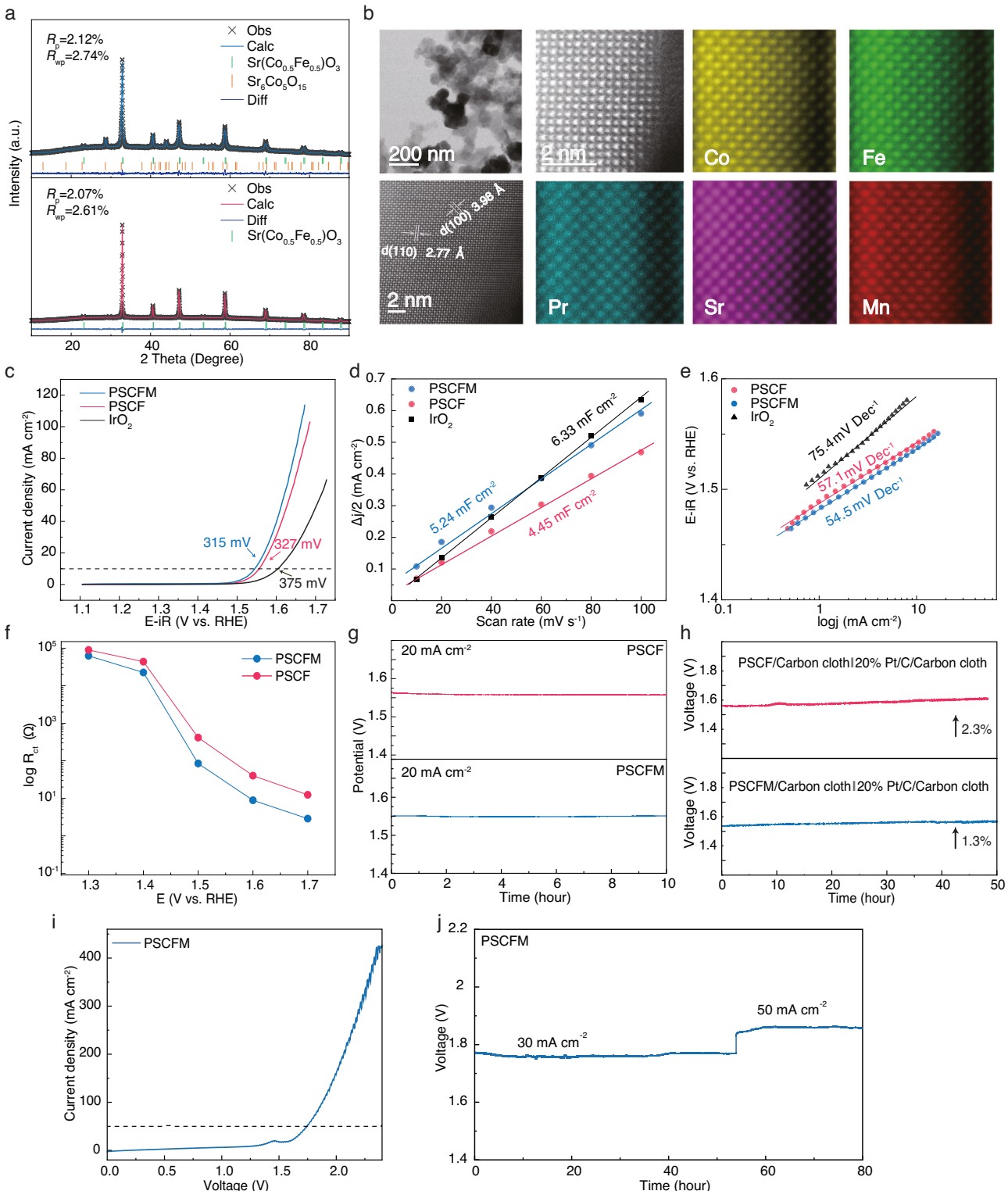

**Fig. 3 | Experimental validation and performance of predicted perovskite electrocatalysts. a** Rietveld refinement of the XRD patterns for PSCF and PSCFM. **b** Representative TEM image, HAADF-STEM image and atomic-scale elemental mapping for Pr, Sr, Co, Fe, and Mn in PSCFM. **c** LSV curves of various electrocatalysts in 1 M KOH electrolyte. The potentials are iR corrected. The measured R for $IrO_2$, PSCF and PSCFM was $5.98 \pm 0.07$, $6.0 \pm 0.04$ and $6.0 \pm 0.06$ Ω, respectively. The curves without iR-correction are shown in Supplementary Fig. 42. **d** Electrochemical double-layer capacitance ($C_{dl}$) plot. **e** Tafel slope plots. The potentials are iR corrected. The measured R for $IrO_2$, PSCF and PSCFM was

$5.98 \pm 0.07$, $6.0 \pm 0.04$ and $6.0 \pm 0.06$ Ω, respectively. **f** Charge transfer resistance ($R_{ct}$) vs. applied voltage without iR-correction. **g** Galvanostatic test of PSCF and PSCFM in a three-electrode configuration at 20 mA cm$^{-2}$ without iR-correction. **h** Galvanostatic test of PSCF and PSCFM in a two-electrode configuration at 10 mA cm$^{-2}$ without iR-correction. 20% Pt/C was used as the cathodic electrode. **i** LSV curve for the electrolyzer in 6 M KOH solution without iR-correction. **j** Stability test of the electrolyzer at current densities of 30 mA cm$^{-2}$ and 50 mA cm$^{-2}$ without iR-correction.

analysis revealed (Supplementary Figs. 20 and 21) an estimated chemical valence of Co is 2.46 and 2.49 for PSCFM and PSCF, respectively. This near-identical chemical valence is confirmed by the Co L-edge XAS (Supplementary Fig. 22), which include reference materials of CoO ($Co^{2+}$) and $Co_3O_4$ ($Co^{2+}/Co^{3+}$) with distinct Co valences. The overlapping Co $L_3$-edge peak positions of PSCF and PSCFM further indicate their similar Co valence states. Consistently, the deconvolution of XPS data indicates that the chemical valence of Fe is 2.38 in PSCF and 2.37 in PSCFM following the incorporation of Mn (valence of 3.39), also conforming with the results of Fe $L_3$-edge XAS data.

Figure 4a compares the O 1s XPS spectra of different samples to distinguish the different surface oxygen species. Deconvolution reveals four well-fitted peaks: lattice oxygen at 528.1 eV (P1), $O^-$ at 529.5 eV (P2), $OH^-/CO_3^{2-}$ at 531.5 eV (P3), and adsorbed water ($H_2O$) at 533.1 eV (P4). Notably, PSCFM exhibited a higher percentage of P2 and P3 (79.6%) compared to PSCF (73.6%), suggesting PSCFM has a higher content of oxygen vacancy-related surface absorbed oxygen species[31–33]. Vacant oxygen sites facilitate nucleophilic attack of $OH^-$ and promote O-O bonding. Previous studies have demonstrated that oxygen vacancies in transition metal oxides induce the formation of new electronic states through hybridization of O $2p$ and metal $3d$ orbitals within the bandgap. These states directly contribute to the enhanced adsorption of intermediates on oxygen vacancies and improved electronic conductivity[34]. In addition, the calculated bandgap for PSCF and PSCFM is 0.61 and 0.41 eV, respectively, consistent

with the resistivity measurements (981 kΩ for PSCF and 387 kΩ for PSCFM at 25 °C). To corroborate the hypothesis of enhanced reactant adsorption, methanol was used as a probe to assess the adsorption capacity of OER intermediates. As $OH^-$ is an electrophilic OER intermediate, it readily reacts with nucleophilic methanol. Consequently, the increase in current density between the methanol oxidation reaction (MOR) and OER polarization curves correlates with the surface coverage of $OH^-$. Prior to analysis, the $C_{dl}$ values for both PSCF and PSCFM catalysts were determined in MOR (Supplementary Fig. 23). The $C_{dl}$ for PSCF (2.15 mF $cm^{-2}$) and PSCFM (2.66 mF $cm^{-2}$) in MOR are similar to those achieved in OER. This indicates that the influence of ECSA on the current increase in MOR is negligible. As shown in Supplementary Fig. 24 and Fig. 4b, the significantly higher MOR current density of PSCFM (2.5 times that of PSCF) clearly demonstrates a stronger affinity for $OH^-$ and thus higher $OH^-$ coverage on the PSCFM surface. Furthermore, to investigate the absorption of reactants on the PSCF and PSCFM during OER, Laviron analysis was conducted for both materials. As shown in Supplementary Fig. 25, the steady-state redox currents associated with $OH^-$ transfer show a linear correlation with the square root of potential scan rates in the cyclic voltammetry (CV, 1 to 35 mV $s^{-1}$) curves. Notably, PSCFM shows a larger redox constant ($K_s = 0.17$ $s^{-1}$) compared to PSCF ($K_s = 0.16$ $s^{-1}$), suggesting a stronger binding strength between *OH intermediates and the surface[35,36]. These results agree with the in situ EIS analysis (Fig. 3f), which suggest enhanced $OH^-$ absorption on the catalyst[37].

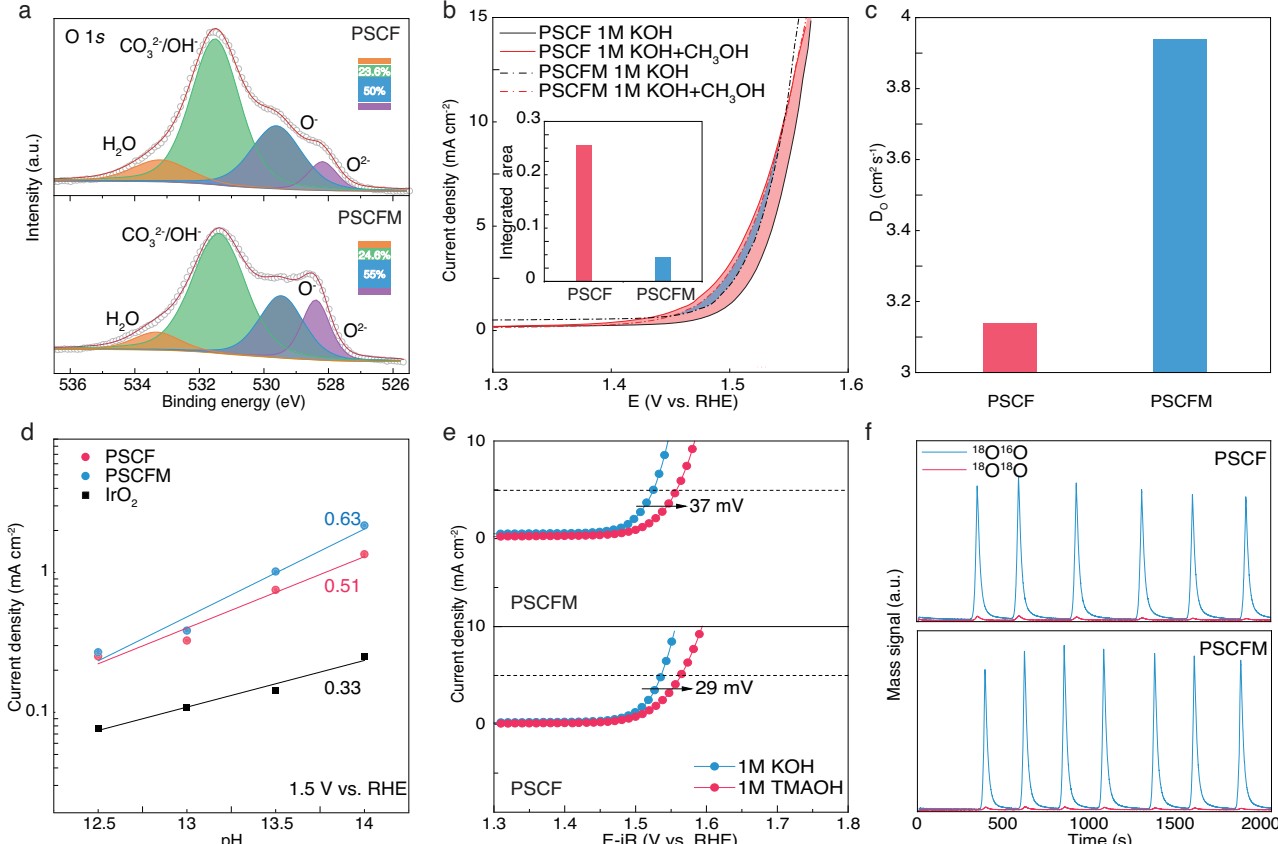

**Fig. 4 | Mechanistic insights into OER for the optimized electrocatalysts.**
**a** Deconvolution of O 1s XPS spectra for PSCF and PSCFM electrodes. The inset figure shows the relative percentage of each fitted peak. **b** LSV curves of PSCF and PSCFM in 1.0 M KOH with and without methanol (0.602 mol $L^{-1}$) without iR-correction. The inset figure compares the integrated area of the current increase region. **c** Comparison of oxygen diffusion coefficients for different electrodes. **d** Current densities of various electrocatalysts at 1.5 V vs RHE under different pH

conditions. **e** Polarization curves of different electrocatalysts in 1 M KOH and 1 M TMAOH electrolytes. The potentials are iR corrected and the measured R for PSCF and PSCFM was 6.0 ± 0.04 and 6.0 ± 0.06 Ω in 1 M KOH, respectively. The potentials are iR corrected and the measured R for PSCF and PSCFM was 3.94 ± 0.16 and 4.01 ± 0.14 Ω in 1 M TMAOH, respectively. **f** DEMS signals for $^{16}O^{18}O$ ($I_{34}$) and $^{18}O^{18}O$ ($I_{36}$) from the reaction products of $^{18}O$-labeled electrocatalysts in 1 M KOH with $H_2^{16}O$.

To further corroborate the high oxygen vacancy concentration in our materials, electrochemical oxygen intercalation in PSCF and PSCFM was examined using CV experiments conducted in an Ar saturated 6 M KOH solution. The observed redox peaks arise from the insertion and extraction of oxygen ions into and from the oxygen-vacant sites. The PSCFM exhibits a larger current density in the intercalation regime, indicating its abundance of sites for oxygen intercalation[38]. Additionally, the oxygen ion diffusion coefficients ($D_o$) of various catalysts were determined using chronoamperometry (Supplementary Fig. 26c, d). Subsequently, we used a bounded three-dimensional (3D) diffusion model, based on Brunauer−Emmett−Teller (BET) surface area (Supplementary Fig. 27), to estimate the $D_o$ of the materials. As shown in Fig. 4c, the $D_o$ of commercial $IrO_2$ is estimated to be $1.34 \times 10^{-12}\,cm^2\,s^{-1}$. PSCF and PSCFM exhibit an order of magnitude higher diffusion coefficient $D_o$ of $3.15 \times 10^{-11}\,cm^2\,s^{-1}$ and $3.95 \times 10^{-11}\,cm^2\,s^{-1}$, respectively, highlighting their fast oxygen exchange capability, as predicted for the electrocatalysts.

## Hybrid reaction mechanism

Preliminary characterizations have collectively confirmed the high concentration of oxygen vacancies in PSCFM and PSCF. These oxygen vacancies could further facilitate the kinetically favorable well-known LOM pathway to bypass the overpotential cap of 370 mV through the adsorbate evolution mechanism (AEM). In the LOM pathway, the deprotonation of hydroxyl groups has become non-concerted and decoupled from subsequent electron transfer[39]. To investigate the feasibility of the LOM pathway, we evaluated the electrocatalysts in a series of KOH electrolytes with varying pH values of 12.5−14 (Supplementary Fig. 28). The pH dependence of OER activity on the reversible hydrogen electrode (RHE) scale indicates the presence of non-concerted proton−electron transfer steps during the OER[40]. It is notable that the increased current density of PSCFM and PSCF at 1.5 V (vs RHE) significantly exceeds that of $IrO_2$ (Fig. 4d), revealing that PSCFM and PSCF exhibit remarkable pH-dependent OER activity and pronounced lattice-oxygen involvement. Moreover, the slopes (ρ) of the OER activity plots on the RHE scale provide insights into reaction orders. The slopes of PSCFM (0.63) and PSCF (0.51) are considerably larger than that of $IrO_2$ (0.33), further confirming the favorable LOM pathway for PSCFM and PSCF during the OER. Furthermore, we compared the OER activities of PSCF and PSCFM electrodes in 1 M KOH and TMAOH solutions. Peroxo-like ($O_2^{2-}$) negative species are generally recognized as key intermediates in the LOM pathway, which can be captured by tetramethylammonium cation ($TMA^+$), leading to retarded OER kinetics for electrocatalysts via the LOM pathway[41]. As depicted in Fig. 4e, the OER activity of PSCFM and PSCF in TMAOH-containing electrolyte is significantly diminished, with overpotential increases of 37 mV and 29 mV, respectively, at a current density of $5\,mA\,cm^{-2}$. This reduction in activity is attributed to the inhibition of the LOM pathway due to the strong binding of $TMA^+$ to peroxo-like ($O_2^{2-}$) negative species, which are key intermediates in the LOM mechanism. Moreover, we performed $^{18}O$ isotope labeling differential electrochemical mass spectrometry (DEMS) to provide direct evidence of lattice oxygen participation during the OER (Fig. 4f). The intense peak corresponding to $^{16}O^{16}O$ (m/z = 32) observed for both PSCF and PSCFM (Supplementary Fig. 29) indicates the presence of oxygen evolution via the AEM, involving the sequential formation of *OH, *O, and OOH intermediates. As shown in Supplementary Fig. 30, in situ attenuated total reflection Fourier transform infrared (ATR-FTIR) spectra of both PSCF and PSCFM catalysts exhibit an absorption band at approximately $1230\,cm^{-1}$, which is characteristic of surface-adsorbed superoxide (*OOH) during the OER[42]. This finding further supports the AEM pathway for both catalysts[43]. Nevertheless, pronounced periodical signals of $^{16}O^{18}O$ (m/z = 34) were detected in both PSCFM and PSCF during the DEMS experiment (Fig. 4f), indicating that the oxygen

atoms in metal-oxygen bonds can also be activated to form O-O bond with neighboring OH or lattice oxygen to release gaseous $O_2$[44]. The calculated contents of $^{18}O^{16}O$ (m/z = 34) for PSCF and PSCFM were around 0.60% and 0.65%, respectively, both higher than the natural isotopic abundance of $^{18}O$ ( ~0.2%) in the electrolyte[45]. Notably, the intensity of the $^{18}O^{18}O$ (m/z = 36) signal for both materials was extremely low, confirming that the released oxygen does not originate from adjacent oxygen atoms in the metal-oxygen bonds. Therefore, the hybrid mechanism of AEM and LOM is confirmed for both electrocatalysts.

To gain further insights into the physical nature of the predicted electrocatalysts for the OER, the oxygen K-edge XAS spectra in total electron yield (TEY) mode were measured for PSCF and PSCFM (Fig. 5a). The spectra collected in TEY mode are surface sensitive (~couple nanometers) due to the limited penetration depth of electrons. The pre-edge peaks below ~530 eV correspond to the oxygen hole states at the conduction band minimum (CBM) induced by the high valent metal[32]. Peaks A and B reflect the degree of hybridization between the oxygen 2p state and the transition metal 3d state[46], while peak C represents the interaction between the oxygen 2p state and the Sr 4d state, and peak D corresponds to the mixed state of the transition metal 4sp orbitals. The normalized intensity and energy position of peaks A and B can be used to characterize the covalent degree of the metal-oxygen bonding, which is a crucial factor influencing oxygen adsorption and redox processes. Clearly, the PSCFM exhibits a pre-edge peak of O K-edge at lower energy compared to PSCF, reducing the energy difference between the redox potentials of $OH/O_2$ and CBM, thereby facilitating electron transfer associated with OER[47]. Moreover, the higher A + B peak intensity of PSCFM compared to PSCF clearly demonstrates the enhanced metal 3d-O 2p hybridization. These experimental findings are supported by theoretical simulation which analyzed the density of state for PSCF and PSCFM to investigate the band gap and rational regulation of transition metal 3d and O 2p orbitals. The overlap of metal 3d and O 2p orbital is further quantified and evaluated by calculating the metal-oxygen covalency, defined as the difference in band center between the metal d-band orbital and oxygen p-band orbitals. As further shown in Supplementary Figs. 31, 32, the metal 3d and O 2p band centers are located at −1.381 eV and −2.871 eV vs. the Fermi level in PSCF. However, their locations shift to −1.318 eV and −2.602 eV after the incorporation of Mn, indicating the increased covalency of metal-O bonds in PSCFM compared to PSCF, thus lowering the energy penalty required for lattice oxygen oxidation.

We next explore the energetic pathway of alkaline OER on both PSCF and PSCFM to rationalize the correlation between high performance and engaged LOM. The evolution of all absorption intermediates via AEM and LOM are presented in Supplementary Figs. 33−36. As shown in Supplementary Fig. 37, the absorption energy of $OH^-$ on Co and Fe cationic sites in PSCF is calculated to be −1.836 eV and −0.471 eV, respectively, indicating that Co sites are the preferable absorption sites. Similarly, Co is also the most energetic favorable absorption site for PSCFM (−2.391 eV on Co, −1.093 eV on Fe and −1.583 eV on Mn). Specifically, in the traditional AEM pathway, the potential determining step (PDS) is the deprotonation of *OOH to form *OO (Intermediate state 3, IS3), which results in a calculated overpotential of 0.71 V on PSCF. We then investigated the energy variation based on LOM. The black curve illustrates the reaction paths for LOM with the release of $^{34}O_2$ (denoted as LOM-$^{34}O_2$). In comparison, in the LOM-$^{34}O_2$ pathway, the PDS is the absorption of OH on $O_v$ (IS3) after $O_2$ release, giving a calculated overpotential of 0.46 V. This DFT calculation strongly suggests that oxygen evolution favors LOM over traditional AEM. After partially substituting Fe with Mn, the calculation of PSCFM further illustrates that the rate determining step (RDS). (IS3) via AEM has a lower calculated overpotential of 0.63 eV. While for LOM, the energy barrier of RDS (IS3) also decreases by 0.12 eV to 0.34 eV. This result is consistent with the experimentally confirmed

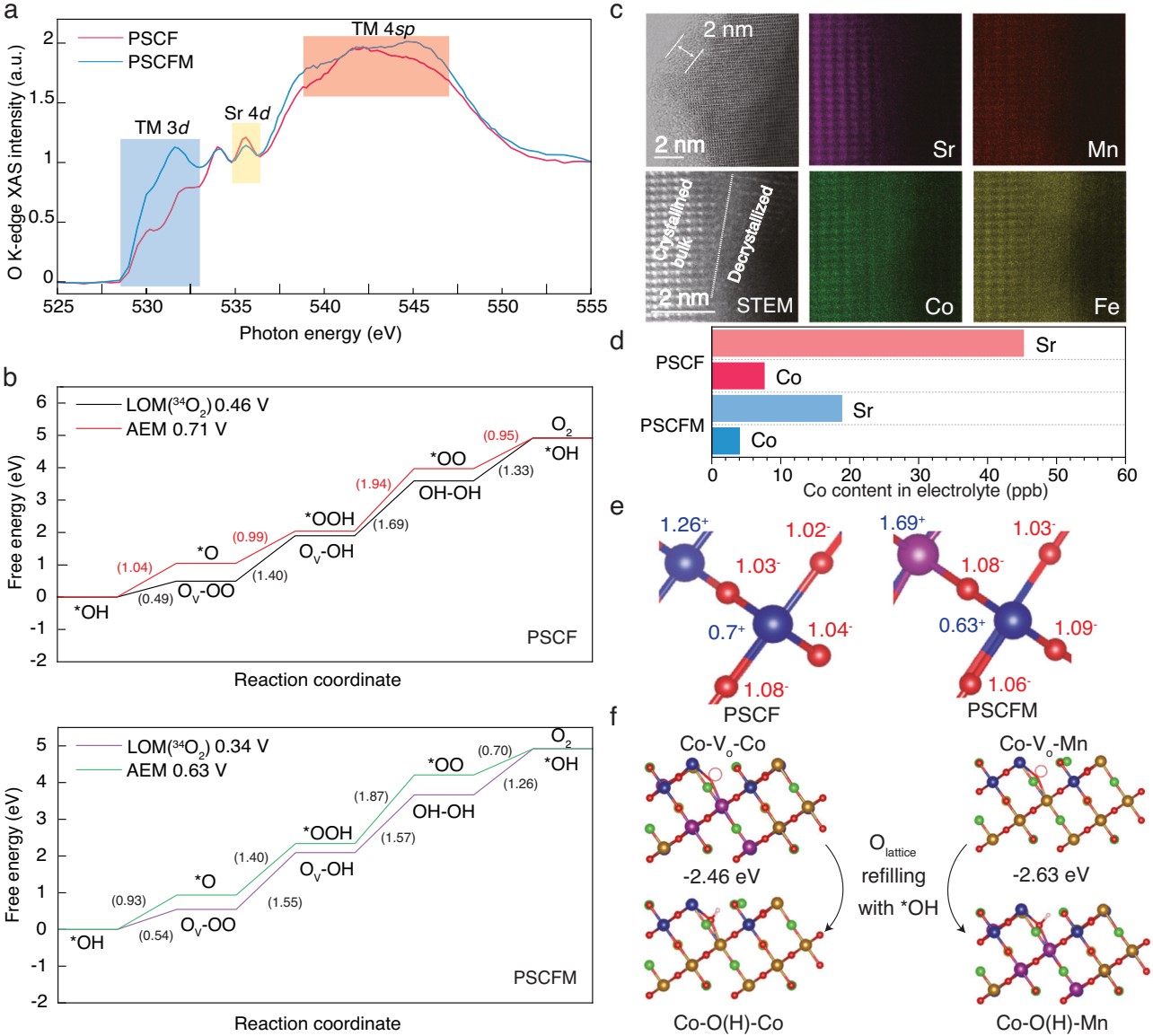

**Fig. 5 | Impact of Mn doping on perovskite oxides for OER. a** O K-edge XAS analysis of PSCF and PSCFM. **b** Free energy diagrams for OER via the AEM and LOM pathways for PSCF and PSCFM. **c** HRTEM image and HAADF-STEM EDX elementary mapping of PSCFM after stability testing. **d** Comparison of Co content in the electrolyte after stability testing, measured by ICP-MS. **e** Oxidation states of PSCF and PSCFM based on Bader charge calculations (electron transfer differences). Blue, red and purple spheres represent Co, O and Mn element, respectively. **f** Pathway and free energy for $V_o$ refilling by absorbed *OH on PSCF and PSCFM. Brown, blue, red, pink, green, purple and yellow spheres represent Fe, Co, O, H, Sr, Mn and Pr element, respectively.

conclusion of facilitated *OH absorption in PSCFM (Fig. 4b). Our theoretical calculation results confirm that the partial substitution of Co with Mn simultaneously reduces the kinetic barrier of AEM and LOM.

## Anti-degradation via Mn-O-Co motif

The PSCFM not only exhibited enhanced electrochemical activity but also demonstrated superior resistance to catalyst decay compared to PSCF, as shown in Fig. 3. To elucidate the origin of this improved stability, the morphology, structure, electronic state, and dissolution rate of the post-OER (after 10 h stability test in two electrode configuration) electrodes were comprehensively characterized to gain insights into the dynamic material evolution during OER. While XRD and SEM data revealed that the overall crystal structures and morphologies of PSCF and PSCFM remained intact after the OER stability test (Supplementary Fig. 38), the high-resolution transmission electron microscopy (HRTEM) image of PSCFM after OER

measurement showed an amorphous surface region with a depth of ~2 nm, adjacent to the bulk with a highly ordered atomic arrangement (Fig. 5c). Atomic scale scanning transmission electron microscopy energy dispersive X-ray mapping further confirmed the B site-rich/A site-deficient nature of the delocalized surface shell. As previously demonstrated, surface amorphization resulting from A-site leaching and B-site redeposition occurs in perovskites with an O 2p-band center closer to $E_F$[48], a characteristic of LOM[31]. Conversely, perovskites with an O 2p band positioned far from the $E_F$ may experience amorphization due to B-site leaching at high potentials. In our study, time-dependent inductively coupled plasma mass spectrometry (ICP-MS) measurements for both samples (Fig. 5d) revealed orders of magnitude higher dissolution rate of Sr than Co in both materials, confirming that A-site leaching is responsible for the reconstruction. Regardless of the underlying mechanism driving the reconstruction, suppressing the dissolution of the crucial B-site element is critical for achieving satisfactory long-term operation.

a

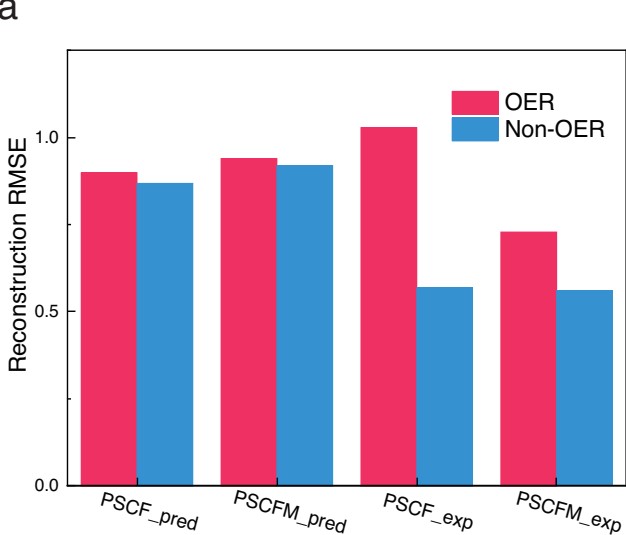

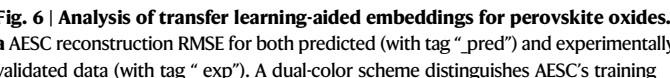

b

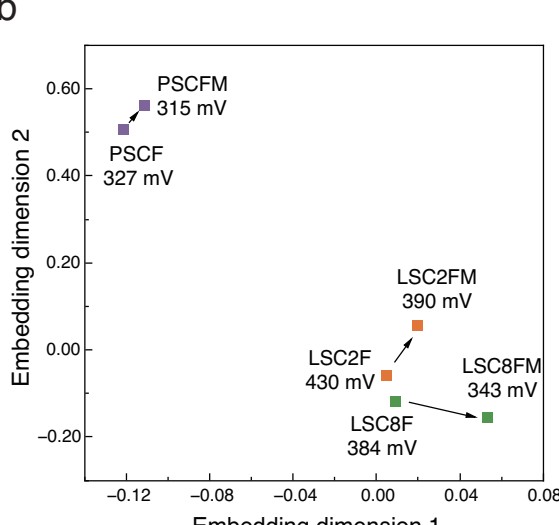

**Fig. 6 | Analysis of transfer learning-aided embeddings for perovskite oxides.** **a** AESC reconstruction RMSE for both predicted (with tag "_pred") and experimentally validated data (with tag "_exp"). A dual-color scheme distinguishes AESC's training dataset. **b** Substructure visualization. This visualization includes pairs of candidate materials with their predicted overpotentials at 10 mA cm$^{-2}$: LSC2F (430 mV) vs. LSC2FM (390 mV), LSC8FM (384 mV) vs. LSC8FM (343 mV), and PSCF vs. PSCFM.

The ICP-MS measurement highlights the higher loss of Co in PSCF (7.59 ppb) compared to PSCFM (4.01 ppb), implying that the formation of Mn-O-Co conjugate is critical for the enhanced stability. To elucidate the mechanism behind its stability, the electron density distribution was investigated. As shown in Fig. 5e, the Bader charges of Co and Mn are positive, indicating their roles as electron donors. The higher electron density on the oxygen bridge $O_{bridge}$ (from 1.03$^-$ to 1.08$^-$) is also evident. Notably, the substation of Co with Mn to form Co-$O_{bridge}$-Mn leads to electron redistribution and electron colocalization around Co, accompanied by a decrease in the Co Bader charge from 0.7$^+$ to 0.63$^+$. This finding aligns with the XPS results. The reduced Bader charge reveals an orderly decrease in the degree of ionization, which can significantly enhance resistance to cation leaching[49].

In addition to the beneficial confinement of Co, the dynamic reversibility of $O_{lattice}/V_O$ is another critical factor influencing surface stability. To complete a reversible LOM cycle, the $V_O$ generated by lattice oxygen migration to form *OOH on M (step 2 in LOM, M is the cation center) should be replenished by further absorption of *OH (step 4 in LOM). However, many electrocatalytic platforms fail to exhibit satisfactory stability due to rapid surface oxygen exchange kinetics[50], resulting in continuous depletion of lattice oxygen. The O 1$s$ XPS data reveal that the percentage of (P2 + P3) peak in PSCF increased by 15.1%, more than twice that of PSCFM (Supplementary Figs. 39 and 40), confirming that lattice oxygen in PSCF is less reversible. These results are further supported by the O K-edge XAS analysis of post-OER electrocatalysts (Supplementary Fig. 41), where a completely diminished O-projected density of unoccupied states was observed in PSCF. Consistent with these experimental findings, the DFT calculations (Fig. 5f) also confirm that the absorption of *OH is thermodynamically more favorable on $V_O$ in the Co-$V_O$-Mn motif (−2.63 eV) than in the Co-$V_O$-Co counterpart (−2.46 eV), facilitating the refilling of $O_{lattice}$ and generation of $BO_6$. Additionally, the Co element exhibits a significant valence increase from 2.49 to 2.59 in PSCF, likely due to substantial surface dissolution and reconstruction (Supplementary Table 9). In contrast, the valence of Co in PSCFM only shows pretty subtle variation (0.01), indicating a more dynamically stable site.

## Discussion

To validate the effectiveness of transfer learning encoding, we employed experimentally characterized PSCF and PSCFM data as benchmarks. As shown in Fig. 6a, incorporating non-OER data resulted in a marginal improvement in reconstruction precision for all predicted PSCF and PSCFM data points with a valence distribution (PSCF: from 0.9 to 0.87; PSCFM: from 0.94 to 0.92). However, for actual experimentally validated perovskite data, the precision of transfer learning-assisted encoding reconstruction exhibited a marked enhancement. The reconstruction RMSE decreased from 1.03 to 0.57 for PSCF and from 0.73 to 0.56 for PSCFM. It is remarkable that both PSCF and PSCFM are novel materials for the AESC, yet the model demonstrates remarkable proficiency in assimilating information pertaining to these uncharted perovskite materials. This highlights the effectiveness of transfer learning in facilitating the discovery of novel materials. Further to PSCF and PSCFM, a comprehensive search of our collected database revealed two additional important data pairs: $La_{0.6}Sr_{0.4}Co_{0.2}Fe_{0.8}$(LSC2F) vs. $La_{0.6}Sr_{0.4}Co_{0.2}Fe_{0.4}Mn_{0.4}$(LSC2FM), $La_{0.6}Sr_{0.4}Co_{0.8}Fe_{0.2}$ (LSC8FM) vs. $La_{0.6}Sr_{0.4}Co_{0.8}Fe_{0.1}Mn_{0.1}$ (LSC8FM). Interestingly, Mn doping in all cases led to enhanced activity. These discoveries suggest potential underlying relationships between the pre- and post-Mn-doped materials that could be exploited to optimize performance.

The concept of vector space substructures, introduced in the context of perovskite oxide embeddings, sheds light on potential direct inter-point connections and deviates from traditional clustering methods by emphasizing relative inter-point relationships rather than dissimilarities and similarities between multiple points[51]. Fundamentally, this structural framework preserves the relative spatial relationships between paired data points sharing similar characteristics. As illustrated in Fig. 6b, these spatial correlations, known as substructures, elucidate to some extent the effects of Mn-doping on the materials, where all points representing Mn-doped materials shift to the right. This finding raises the possibility that leveraging these substructures could facilitate the inference of spatial orientations for other materials post-Mn-doping, paving the way for the prediction and enhancement of their functional properties. Moreover, this discovery presents a novel paradigm for material optimization through the integration of experimental methodologies and ML.

Despite recent advancements, active learning methodologies remain crucial for perpetually expanding the explorable space in material discovery. This principle aligns with the recent findings by Merchant et al., who demonstrated that expanded dataset enhances

model performance in material science[52]. Similarly, our experimental validation of PSCF and PSCFM confirms their systematic identification of superior perovskites. Furthermore, the accuracy of these predictions progressively improves with each iteration. Moving forward, we aim to significantly augment the current dataset, further enhancing the framework's stability. Additionally, we plan to incorporate models with stronger interpretability to gain deeper insights into the OER process.

Leveraging polyphyletic data sources, we established a transfer ML framework that exhibits enhanced generalization capabilities for the precise estimation of a family of non-noble-metal perovskite oxide electrocatalysts for alkaline OER. The global ensemble model demonstrated attractive predictive accuracy, which was validated through experimental data, confirming the model's ability to forecast the properties of novel materials. By incorporating active learning techniques, we successfully expanded the predictive scope to encompass more complex systems. Notably, the PSCF and PSCFM electrocatalysts exhibited low overpotentials of 327 mV and 315 mV at 10 mA cm$^{-2}$, respectively. A comprehensive suite of in situ/operando characterization techniques revealed that both AEM and LOM pathways operate on the electrocatalysts, and PSCFM, with enhanced OH$^-$ absorption and richer surface oxygen vacancies, is more inclined towards the O$_{lattice}$-O$_{OH}$ coupling. The DFT calculations demonstrated strengthened metal 3$d$-O 2$p$ covalency and a lower energy cost for OH$^-$ absorption on O$_v$ in PSCFM, unlocking a kinetically faster LOM pathway beyond the overpotential ceiling of AEM. More significantly, Mn incorporation was found to decelerate the dissolution of the reactive Co center by redistributing charge density and expedite the reversibility of O$_{lattice}$/V$_O$ during LOM. This work establishes a powerful ML paradigm and provides mechanistic insights, paving the way for accelerating the development of high-performance OER catalysts.

# Methods

## Chemicals

Barium nitrate (Ba(NO$_3$)$_2$, 99.99% metals basis), copper nitrate trihydrate (Cu(NO$_3$)$_2$·3H$_2$O, AR), nickel nitrate hexahydrate (NiN$_2$O$_6$·6H$_2$O, AR), niobium(V) oxalate hydrate (C$_{10}$H$_5$NbO$_{20}$·xH$_2$O, 98%), zirconium nitrate pentahydrate (Zr(NO$_3$)$_4$·5H$_2$O, 99.9%), stannic chloride hydrated (SnCl$_4$·5H$_2$O, AR), iridium (IV) chloride (IrCl$_4$·xH$_2$O, ≥99.9% metals basis), ruthenium chloride hydrate (RuCl$_3$·xH$_2$O, 99.95% metals basis), methanol (CH$_3$OH, AR), tetramethylammonium hydroxide solution (TMAOH, AR), water-$^{18}$O (H$_2$$^{18}$O, 97 atom % $^{18}$O) were purchased from Shanghai Aladdin Bio-Chem Technology Co., Ltd. Cobalt nitrate hexahydrate (Co(NO$_3$)$_2$·6H$_2$O, 99.99% metals basis), manganese acetate (C$_4$H$_6$MnO$_4$, AR), citric acid (CA, ≥99.5%), potassium hydroxide (KOH, AR), anhydrous ethanol (AE, 99.5%) were provided by Shanghai Macklin Biochemical Co., Ltd. Praseodymium nitrate hexahydrate (PrN$_3$O$_9$·6H$_2$O, 99.99%), strontium nitrate (Sr(NO$_3$)$_2$, AR), iron nitrate nonahydrate (FeN$_3$O$_9$·9H$_2$O, 99%), calcium nitrate tetrahydrate (Ca(NO$_3$)$_2$·4H$_2$O, AR), lanthanum nitrate hexahydrate (LaN$_3$O$_9$·6H$_2$O, AR), ethylenediaminetetraacetic acid (EDTA, AR), ammonia solution (NH$_3$·H$_2$O, GR) were supplied by Sinopharm Chemical Reagent Co., Ltd. Iridium oxide (IrO$_2$, SINY85) and platinum carbon catalyst (20 wt% Pt/C, TANAKA) were purchased from Sinero.

## Material synthesis

The PSCF, PSCFM and other catalysts were synthesized using a sol-gel method. As an example, the typical process for synthesizing PSCF is detailed as follows: 0.2 mmol of Pr(NO$_3$)$_3$·6H$_2$O, 1.8 mmol of Sr(NO$_3$)$_2$, 1 mmol of Co(NO$_3$)$_2$·6H$_2$O, 1 mmol of Fe(NO$_3$)$_3$·9H$_2$O and 4 mmol of CA were dissolved in 30 ml of H$_2$O. Then, 4 mmol of EDTA was slowly added. Finally, the NH$_3$·H$_2$O was dropwise added to adjust the pH value to 7–8. The beaker containing the solution was then placed on a heating platform at 90 °C with continuously stirring until a gel was formed. The beaker was then transferred to an oven at 200 °C for 10 h. After that, the obtained precursor was taken out, thoroughly ground,

and then pre-calcined in a muffle furnace at 400 °C for 2 h. Subsequently, the sample was heated to 850 °C at a heating rate of 5 °C min$^{-1}$ and maintained at 850 °C for 6 h to obtain PSCF powder. For the synthesis of PSCFM, the stoichiometric amount of C$_4$H$_6$MnO$_4$ was used as a precursor to replace the stoichiometric amount of Fe.

## Catalyst characterizations

XRD was performed using an Ultima IV powder diffractometer (Rigaku Corporation, Japan) with Cu K$_\alpha$ radiation (λ = 0.15406 nm), and Rietveld refinement was carried out using the EXPGUI GSAS software. SEM images were obtained using a Zeiss GeminiSEM 500 instrument. HRTEM, HAADF-STEM and elementary distribution mapping were performed using a JEOL-ARM 300 F instrument operated at an accelerated voltage of 300 kV. The chemical state of the materials was analyzed by XPS using a Thermo Scientific K-Alpha instrument (Al K$_\alpha$ radiation = 1486.6 eV). The binding energy of C 1$s$ (284.8 eV) was used for calibration. The conductivity of the materials was measured at room temperature using a ST2643 Ultra-high Resistance Micro-current Tester. The BET surface areas were determined by N$_2$ adsorption−desorption isotherms at −196 °C using an ASAP 2020 instrument. K-edge XAS of O and L-edge XAS of Co, Fe and Mn were measured at the National Synchrotron Radiation Laboratory (China) in total electron yield mode at high vacuum (~10$^{-5}$ mbar). The dissolution of catalysts during electrolysis was quantified by inductively coupled plasma mass spectrometry (Thermo Fisher), with a volume (10 mL) of electrolyte used to meet the detection limit of the equipment.

## Electrocatalytic activity for OER

A 5 mm diameter rotating disk electrode (E5 RDE, PINE) was used to evaluate the electrocatalytic properties of the catalysts. Cyclic voltammetry, LSV, EIS, chronopotentiometry and other electrochemical tests were performed using an electrochemical workstation (Corrtest 2350H). To prepare the catalyst ink, 10 mg of catalyst powder and 2 mg of acetylene black were mixed in a container. Then, 470 µL of anhydrous ethanol, 500 µL of deionized water and 30 µL of Nafion solution (DUPont D520, 5 wt%) were pipetted into the container for dissolution. The mixture was sonicated in an ice bath for more than 2 h to ensure even mixing and dispersion of the catalyst and acetylene black. The rotating disc electrode was ground with a suede soaked in alumina suspension. Subsequently, a 10 µL aliquot of the ink was pipetted onto the RDE and dried at room temperature. This resulted in a catalyst mass loading of 0.51 mg cm$^{-2}$.

A traditional three-electrode system was used for electrochemical testing in a 150 mL water-bath electrolytic cell containing 1 M KOH saturated with oxygen (99.99%). A Hg/HgO electrode (calibrated with an RHE in 1 M KOH) served as the reference electrode, while a graphite rod functioned as the counter electrode. The electrolytes were freshly prepared by dissolving a stoichiometric amount of KOH in ultrapure water (18.2 MΩ cm$^{-1}$) and stored in volumetric flasks. The catalyst activity for the OER was evaluated by LSV (from 1.2 to 1.8 V vs RHE at 10 mV s$^{-1}$). Before testing, the working electrode was scanned for 10 cycles (1.2–1.8 V vs RHE) to achieve a stable cyclic voltammogram. The Tafel curves were obtained at a slow scan rate of 1 mV s$^{-1}$. The Hg/HgO reference electrode calibration involved two Pt electrodes initially cycled 45 times in 0.5 M H$_2$SO$_4$ for surface cleaning (to remove contaminants) between −2 and 2 V at 50 mV s$^{-1}$ for 2 h. These Pt electrodes were then used as the working and counter electrodes, respectively. Prior to calibration, pure hydrogen was bubbled through the 1 M KOH for at least 30 min to ensure proper saturation. The LSV curve was recorded at 1 mV s$^{-1}$ to identify the potential of zero net current (−0.905 V, as shown in Supplementary Fig. 43). Based on the Nernst equation, all potentials in this work were reported versus RHE in 1 M KOH (pH = 14) using the following equation: E$_{vs RHE}$ = E$_{vs Hg/HgO}$ + 0.905 V. The EIS was also performed at various potentials with a frequency range of 10$^5$ Hz to 10$^{-1}$ Hz. The resistance of electrolyte was

determined by the high-frequency intercept of the EIS plot on the X-axis. The TOF of all catalysts was calculated using the following Equation:

$$TOF = 4 \times j \times s / (n \times F) \tag{1}$$

where $j$ is the measured current density, $s$ is the electrode area, $F$ is the Faraday constant, and $n$ is the molar number of active sites per unit area. It was assumed that all transition metals in the B-site could serve as reactive sites.

The ECSA of each catalyst was estimated based on the electrochemical double-layer capacitance ($C_{dl}$) of the electrocatalyst. The specific value of $C_{dl}$ was calculated by linear fitting the plots of current density difference ($\Delta j/2$) versus scanning rates obtained from CV. The ECSA was obtained using the following equation[53]:

$$ESCA = \frac{C_{dl}}{C_s} (C_s = 0.04 \, mF \, cm^{-2}) \tag{2}$$

Here, $\Delta j/2$ represents half the difference in current density between the positive and negative sweeps of the CV curve at a specific potential.

Before the in situ EIS tests, CV measurements were performed on the working electrode at a scanning rate of 100 mV s$^{-1}$ between 1.1 and 1.8 V vs RHE to activate the electrocatalysts and obtain a stable cyclic voltammogram. The EIS measurements were conducted in the potential range of 1.3–1.7 V versus RHE with a frequency range of 10$^5$ Hz to 10$^{-1}$ Hz.

To evaluate the stability of the catalysts, two methods were employed: a two-electrode system with chronopotentiometry (CP) and a three-electrode configuration with CP. In both methods, the catalyst ink was loaded onto carbon cloth with a mass loading of 1 mg cm$^{-2}$. For the two-electrode setup, a commercially available 20 wt% Pt/C electrode served as the hydrogen evolution electrode to ensure efficient hydrogen evolution and maintain cell performance. The catalyst-loaded carbon cloth functioned as the oxygen evolution electrode. In the three-electrode configuration, a Hg/HgO electrode and a graphite rod were used as the reference and counter electrodes, respectively. The stability test was conducted in 1 M KOH solution within a sealed 200 mL electrolytic cell.

A zero-gap alkaline water electrolyzer was assembled using PSCFM/NF and Pt/C/NF as the anode and cathode, respectively (NF refers to nickel foam). Electrochemical measurements were performed at room temperature using a CHI 760E electrochemical workstation (Chenhua, Shanghai). The catalyst ink was prepared by dispersing 10 mg of catalyst powder in 1 mL of anhydrous ethanol and 50 µL of anion exchange ionomer (Fumion FAA-3-SOLUT-10, FuMA-Tech) using sonication. The catalyst ink was then drop-coated onto the surface of the NF electrodes to achieve desired mass loadings of 1 mg cm$^{-2}$ and 3 mg cm$^{-2}$. The two electrodes were separated by a pre-treated alkaline exchange membrane (Fumasep FAA-3-30, thickness ~30 µm, FuMA-Tech), Prior to use, the membrane was immersed in 6 M KOH solution for 12 h. LSV measurement was performed at a scan rate of 10 mV s$^{-1}$ from 0 to 2.4 V to evaluate cell performance. To assess the long-term stability, CP tests were carried out at constant current densities of 30 mA cm$^{-2}$ and 50 mA cm$^{-2}$, respectively.

The oxygen diffusion coefficient of the electrocatalysts was measured in an Ar-saturated 6 M KOH electrolyte using a three-electrode configuration[31]. The working electrode was the as-prepared electrocatalyst, while a Hg/HgO electrode and a graphite rod served as the reference and counter electrodes, respectively. CV scans were performed at a fixed potential range of 0.6 to 1.4 V vs. RHE with a scanning rate of 20 mV s$^{-1}$ to characterize the electrode. The chronoamperometry method was used to measure the oxygen ion diffusion coefficient of the electrocatalysts. During the timed current collection process, the potentials applied to PSCF and PSCFM electrodes were

0.167 and 0.20 V vs. Hg/HgO, respectively. The intersection point between the fitted line of the measured data and the x-axis was determined by plotting the reciprocal square root of current and time ($i$ vs $t^{-1/2}$). According to the bounded 3D diffusion model, the oxygen diffusion coefficient was calculated using the following equation:

$$\lambda = a \times (D_o \times t)^{-1/2} \tag{3}$$

where $\lambda$ is a dimensionless shape factor of 2 and $D_o$ is the oxygen diffusion coefficient. The value of '$a$' is calculated using the following equation:

$$S = 6 / (2a \times \rho) \tag{4}$$

where $S$ is the BET area of the electrocatalysts and $\rho$ is theoretical density of the material.

The MOR tests used an electrolyte containing 1 M KOH with methanol (0.602 mol L$^{-1}$). The resting procedure remained identical to the standard OER measurement.

The DEMS measurements were conducted using a QAS 100 mass spectrometer with a detector (Linglu Instruments, Shanghai) to identify the source of oxygen in the evolved gas during MOR. The catalyst ink was prepared by sonicating a mixture of 3 mg catalyst, 500 µL deionized water, 460 µL absolute ethanol, and 40 µL of 5 wt% Nafion for 1.5 h in an ice bath. Before measurement, 30 µL of the well-dispersed ink was drop-coated onto a gold film electrodeposited on a polytetrafluoroethylene (PTFE) membrane (porosity ≥50%, aperture ≤20 nm, thickness ~40 µm, Linglu, Shanghai) pre-labeled with $^{18}$O in a 1 M KOH electrolyte containing H$_2$$^{18}$O.

The electrocatalyst labeled with $^{18}$O was used as the working electrode. An Ag/AgCl electrode and a Pt wire were used as the reference and counter electrodes, respectively, as illustrated in Supplementary Fig. 44, respectively. The potentials were measured relative to the Ag/AgCl reference electrode and converted to the RHE scale in 1 M KOH (pH = 14) using the following equation: $E_{vs \, RHE} = E_{vs \, Ag/AgCl} + 1.023$ V. Electrochemical measurements were performed at room temperature in a standard DEMS electrochemical cell using a CHI 760E electrochemical workstation (Chenhua, Shanghai). The typical LSV scans were carried out from 0 to 0.65 V vs. Ag/AgCl at a slow scan rate of 5 mV s$^{-1}$ in a 1 M KOH electrolyte containing H$_2$$^{16}$O. Simultaneously, on-line mass spectrometry monitored the electrolyzed gas products for different molecular weights: 32 ($^{16}$O$^{16}$O), 34 ($^{16}$O$^{18}$O), and 36($^{18}$O$^{18}$O).

The Faraday efficiency (FE) of the OER was determined in a 10 mL miniature, two-compartment electrolytic cell separated by a Nafion-N117 membrane (thickness ~183 µm)[54]. A carbon paper working electrode loaded with 1 mg cm$^{-2}$ of catalyst was used. An Ag/AgCl electrode served as the reference electrode, and a Pt wire was used as the counter electrode. The oxygen concentration (c) was calculated using the following equation based on the GC peak areas ($S$):

$$c = 21\% \times \frac{S \times p2}{S \times p1} \tag{5}$$

where p1 is the contrast signal of oxygen in air (21%), and p2 is the oxygen signal produced by the OER. Prior to the reaction, 1 M KOH solution was purged with He gas. The electrolytic water reaction was then carried out at 0.1 A. The generated gas was transported to the anode chamber by He gas at a rate of 20 mL min$^{-1}$, and then discharged into the gas chromatograph to obtain the peak signal. Based on this, the FE can be calculated using the following equation:

$$FE = \frac{4c \times V \times F}{V_m \times I \times t} \tag{6}$$

where $V$ is the volume of the collected gas, $V_m$ is the molar volume constant of the gas, $F$ is the Faraday's constant, $I$ is the application constant current and $t$ is the time. It is important to note that the measurements were only performed once.

The redox constant ($K_s$) of catalysts was calculated using Laviron Equation as follows:

$$E_C = E_{1/2} - (RT/\alpha nF) \times \{\ln(\alpha nF/RTK_s) + \ln(\nu)\} \tag{7}$$

where $E_c$ is the reduction potential of metal redox, $E_{1/2}$ is the formal potential of metal redox, R is the universal gas constant, $T$ is the temperature in kelvin, $n$ is the number of electrons transferred, $\alpha$ is the transfer coefficient, $K_s$ is the rate constant of metal redox, and $\nu$ is the scan rate in the CV measurements[55].

### In situ electrochemical attenuation total reflection Fourier transform infrared spectroscopy (EC-ATR-FTIRS) measurement

To prepare the catalyst electrode, a catalyst ink was first prepared by sonicating a mixture of 10 mg catalyst, 10 mg acetylene black, 1 mL of absolute ethanol, and 100 μL of 5 wt% Nafion for 1.5 h in an ice bath. This ink was then dropped onto a hemicylindrical silicon prism pre-coated with a gold layer. The catalyst electrode, along with a Pt foil counter electrode and a Hg/HgO reference electrode (as shown in Supplementary Fig. 45), were used in a three-electrode configuration filled with 1 M KOH electrolyte. The electrode potential was systematically varied from 1.45 V to 1.85 V in steps. At each potential step, an infrared spectrum was collected with a time resolution of 10 s per scan. The spectroscopy measurements were performed at the National Synchrotron Radiation Center (China).

### Computational methods

The Vienna Ab Initio Package (VASP)[56,57] was used for DFT calculations within the generalized gradient approximation (GGA) with the PBE functional[58] formulation. Project augmented wave (PAW) potentials[59,60] were used to describe the ionic cores, with a plane-wave basis set and a kinetic energy cutoff of 520 eV to account for valence electrons. Partial occupancies of the Kohn-Sham orbitals were allowed using the Gaussian smearing method with a width of 0.05 eV. Convergence criteria for energy and force were set to $10^{-5}$ eV and 0.05 eV Å$^{-1}$, respectively. Grimme's DFT-D3 methodology[61] was used to account for dispersion interactions. The equilibrium lattice constants of $Pr_{0.1}Sr_{0.9}Co_{0.5}Fe_{0.5}O_3$ and $Pr_{0.1}Sr_{0.9}Co_{0.5}Fe_{0.3}Mn_{0.2}O_3$ unit cells were optimized using a $3 \times 3 \times 2$ Monkhorst-Pack k-point grid for Brillouin zone sampling. Subsequently, a surface model with p ($2 \times 2$) periodicity in the x and y directions, and a 18 Å vacuum layer along the c-axis was constructed to represent the surface slab. During structural optimizations for the surface model, only the Γ point in the Brillouin zone was used for k-point sampling, and all atoms were allowed to relax. The free energy of gas phase molecules or adsorbates on the surface was calculated using the equation: G = E + ZPE − TS, where E is the total energy obtained from the calculations, ZPE is the zero-point energy, T is the temperature (set to 298.15 K here), and S is the entropy. This approach was used to determine the sub-reaction free energies (ΔG) for both the AEM and the LOM pathways. Finally, the elementary reaction with the highest free energy barrier was identified as the rate-determining step (RDS).

### Data availability

All data supporting our findings are available in this paper and its associated supplementary materials (Supplementary Information and Supplementary Data 1). Source data for the figures, including Supplementary figures, can be found in the Source Data file. Source data are provided with this paper.

### Code availability

The codes used in this study have been deposited in the GitHub repository and can be found with the following link: https://github.com/helaoer/Transfer-learning-guided-the-discovery-of-efficient-perovskite-oxide-for-alkaline-water-oxidation.

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

## Acknowledgements

This work was supported by the National Natural Science Foundation of China (nos. 22272136, 22102135, 52272233, 22209072). Y.S. acknowledges the financial support from Guangdong Basic and Applied Basic Research Foundation (2022A1515010069), the Science and Technology Project of Fujian Province (2022L3077), and Shenzhen Science and Technology Program (JCYJ20220530143401002). X.T. acknowledges the support from the Engineering and Physical Sciences Research Council (EP/V036696). Y.Z. acknowledges the financial support from the Natural Science Foundation for Young Scholars of Jiangsu Province (no. BK20220879).

## Author contributions

C.J., H.H., H.G., X.T., and Y.S. conceived the idea. C.J., H.G., and Y.S. designed the experiments. H.H. designed the machine learning framework and wrote the code. C.J. synthesized the materials. C.J., X.Z., Q.H., Y.W., and Y.Z. performed material characterizations and electrochemical measurements. X.F. performed the DFT calculations. X.T. and Y.S. supervised the project. C.J., H.H., N.Y., X.T., and Y.S. co-wrote the manuscript. All authors commented on the manuscript.

## Competing interests

The authors declare no competing interests.
