## [Peer Review File · Nature Communications]

Transfer learning guided discovery of efficient perovskite oxide for alkaline water oxidationREVIEWER COMMENTS

Reviewer #1 (Remarks to the Author):

In this manuscript, a transfer machine learning framework is developed to estimate non-noble-metal perovskite oxide electrocatalysts in alkaline OER. The prediction accuracy of this model has been successfully verified by experiments, and two multi-component perovskites have been screened as excellent OER catalysts. This work may be a paradigm for the successful application of machine learning to OER catalyst screening and is expected to help accelerate the development of high-performance catalysts. Therefore, I recommend its publication after addressing the following comments.

1. The experimental apparent activity of OER catalyst is affected by many factors, such as test conditions, catalyst morphology, synthesis parameters, etc. Then how to ensure that the model established based on such highly uncertain data is reliable? In addition, the specific experimental data referenced do not appear to be available.
2. Can the catalysts with the best predicted activity (especially those with an overpotential of less than 300mV) be verified experimentally?
3. EPR results should be provided to demonstrate the presence of high oxygen vacancy concentration in PSCF and PSCFM.
4. The theoretical activities of Co, Fe and Mn sites in PSCF and PSCFM should be carefully compared.
5. The calculation paths of AEM and LOM seem to be unusual. In common AEM calculations, the initial model is always an unsaturated metal site. In the LOM path, some paths are merged, for example from *OH to Ov-OO, involving both deprotonation and oxygen migration, and the OH-OH to O₂ path is confusing. Please provide the basis for the calculation paths.

Reviewer #2 (Remarks to the Author):

This manuscript reported a novel transfer learning paradigm centered on the cation information to predict undiscovered perovskite oxides for alkaline OER. 13 candidates of representative perovskite oxides were identified and synthesized to form pure perovskite structures. The as-synthesized Pr_{0.1}Sr_{0.9}Co_{0.5}Fe_{0.5}O₃ (PSCF) and Pr_{0.1}Sr_{0.9}Co_{0.5}Fe_{0.3}Mn_{0.2}O₃ (PSCFM) exhibited low OER activity with overpotentials of 327 mV and 315 mV at 10 mA cm⁻², respectively. The authors performed in situ/operando measurements and DFT calculations to confirm the different oxide path mechanisms between the PSCF and PSCFM catalysts. However, lots of conclusions of this work were made without solid experimental supports. Therefore, this manuscript is recommended for publication on Nature Communications after addressing the following issues:

1. To demonstrate the advantage of four and five metals, the authors need to prepare three and two metals-based catalysts for comparison.
2. What were the contents of Pr, Sr, Co, Fe, Mn and O elements in the as-synthesized materials? The authors only used XPS to characterize the catalysts. However, XPS is a surface sensitive technique. Therefore, the EDS spectra should be given.
3. The Faradic efficiency during OER should be provided.
4. The stability test with a current density of 20 mA cm⁻² and a testing time of 10 h was far not enough. Only using such lower current density is not meaningful. Larger current densities and much longer stability test close to industrial application should be performed to further illustrate the reveal application potential of this catalyst.
5. Characterizing the oxygen vacancies using O 1s XPS data was not rigorous. The authors

may refer to the following references: Surface Science, 2021, 712, 121894. Since V_o plays an important role in regulate the OER activity, can the authors tune the concentration of V_o ?

6. The authors focus on the role of Mn and Co elements in oxide path mechanisms. What role do the other three metals play?

7. The authors should evaluate the performance of the as-synthesized PSCFM catalyst in anion exchange membrane (AEM) electrolyzer.

Reviewer #3 (Remarks to the Author):

This work reported a transfer learning paradigm to predict perovskite oxides for oxygen evolution reaction. The authors found that $\text{Pr}_{0.1}\text{Sr}_{0.9}\text{Co}_{0.5}\text{Fe}_{0.5}\text{O}_3$ (PSCF) and $\text{Pr}_{0.1}\text{Sr}_{0.9}\text{Co}_{0.5}\text{Fe}_{0.3}\text{Mn}_{0.2}\text{O}_3$ (PSCFM) exhibited good OER activity. They attributed this to the incorporation of Mn into PSCF strengthened the stability of Co reactive sites and lowered the reaction barrier. Electrochemical measurements revealed the coexistence of the adsorbent evolution mechanism and lattice oxygen mechanism for O-O coupling. The main drawback of this study is that the structure of catalysts is not well understood. In addition, several important issues need to be addressed:

(i) The authors stated a transfer learning paradigm. What is the major difference between it and density functional theory calculations and machine learning? What are the key parameters that determine the prediction results?

(ii) The authors stated that the enhanced OH^- absorption and surface oxygen vacancies played a critical role in promoting O-O coupling during the OER process. However, more evidence for the strength of OH^- absorption and concentration of surface oxygen vacancies should be provided. Some key spectra characterizations are missing.

(iii) The authors demonstrated that the incorporation of Mn into PSCF to form Co-Obridge-Mn motifs strengthened the stability of CO reactive sites and lowered the reaction barrier. It is worth noting that the chemical valence of Co is 2.46 and 2.49 for PSCFM and PSCF based on the XPS results. How to understand the interaction between Co and Mn?

(iv) Based on the characterization and DFT calculations, it was suggested that Co was the active site, and the introduction of Mn improved the stability of Co centers. However, Fe is also important for the adsorption and activation of oxygen species. Is there any evidence to exclude this? Confirmation of active center needs additional evidence.

(v) The authors suggested that the catalysts showed good electrolysis durability. While the structure of catalysts after the stability test was missing. More evidence for the stability of PSCF and PSCFM should be provided and analyzed the samples after the reaction.

Response Letter

We have highlighted the revised portions of the manuscript in blue for your reference. We hope the revised manuscript is now suitable for publication in Nature Communications.

Reviewer #1 (Remarks to the Author):

In this manuscript, a transfer machine learning framework is developed to estimate non-noble-metal perovskite oxide electrocatalysts in alkaline OER. The prediction accuracy of this model has been successfully verified by experiments, and two multi-component perovskites have been screened as excellent OER catalysts. This work may be a paradigm for the successful application of machine learning to OER catalyst screening and is expected to help accelerate the development of high-performance catalysts. Therefore, I recommend its publication after addressing the following comments.

Response:

Thanks for your comment. We are grateful for your positive comments and appreciate all the valuable suggestions and questions in the meantime. We have addressed the recommended revisions in the manuscript and provided a point-to-point response shown as follows. We believe these improvements enhance the overall quality of the manuscript.

1. The experimental apparent activity of OER catalyst is affected by many factors, such as test conditions, catalyst morphology, synthesis parameters, etc. Then how to ensure that the model established based on such highly uncertain data is reliable? In addition, the specific experimental data referenced do not appear to be available.

Response: Thanks for your comment.

Data uncertainty. While predicting the OER performance of undiscovered perovskite electrocatalysts, our approach not only encompasses the analysis of the compositional information of the materials, but also incorporates the measurement conditions for OER. The following content in the “Data extraction and cation encoding” section of manuscript has illustrated the related details. Also, most materials we collected were prepared using the sol-gel method. We incorporated the related description in the revision.

Revision in main text

The OER-specific subset is more comprehensive than the non-OER subset, encompassing several characteristic measurement parameters of OER, including mass loading (in mg cm^{-2}), electrolyte (in mol L^{-1}), substrate type (rotating disk electrode, glassy carbon or classy carbon electrode), and overpotential to reach 10 mA cm^{-2} (in mV). Most samples were synthesized using the sol-gel method.

Given the considerably diverse descriptors of data from different literature sources, as well as the

variations in the description of methodologies, it is imperative to streamline and standardize the data, which aims to enhance data consistency and optimize the foundation for subsequent machine learning (ML) training processes. Wang et al. tried to ensure data consistency by segmenting the gathered data points into multiple subsets for training purposes (*J. Am. Chem. Soc.* **2023**, 145, 20, 11457–11465). In our case, instead of categorizing the diverse reaction conditions into subsets, we incorporated them as input dimensions for the model training process. This strategy significantly enhances the volume of data available for utilization. We also restricted our analysis to data points where the electrolyte concentration is 0.1/1 M KOH, the temperature is room temperature, and the synthesis approach is the sol-gel method. For data points that did not show the mass loading, we solely employed them for encoding the material information. This data-processing methodology substantially enhances the consistency of the data available for analysis, thus improve the reliability and reduce uncertainty of the system. We also did modification in the manuscript to better clarify this point.

Revision in main text

We subsequently employed the embeddings of OER materials, along with their corresponding reaction conditions, as input variables for the model. 91 out of a total of 94 points were included, and the rest 3 points were excluded due to their uncertainty in the reaction conditions. The model was then trained to predict the overpotential at a current density of 10 mA cm⁻².

Prediction uncertainty. Furthermore, given the extensive scope of the exploratory space, we adopt a guessed-state distribution approach to generate new data points during the screening of novel materials. A single perovskite formula can correspond to an immense quantity of data points. We assess the uncertainty of prediction by evaluating the predictive values derived from these non-directly corresponding data points. As evidenced in the third predictive iteration for PSCFM (Figure 2h), a reduction in predictive variance signifies a diminution in the uncertainty associated with the predictive model.

Despite these advancements, the need for active learning methodologies to continuously broaden the scope of explorable space remains paramount. Merchant et al. have established that the augmentation of material-related information leads to an enhancement of the model's predictive ability, akin to the scaling law observed in natural language processing (NLP) (*Nature* **2023**, 624, 80–85). In our case, experimental validation of PSCF and PSCFM has confirmed that this method systematically discovered perovskites with superior performance. Furthermore, the accuracy of its predictions enhances progressively with each iteration. In our forthcoming endeavors, we aim to substantially augment the current dataset, thereby elevating the overall stability of the prediction framework. We also added more clarification of this point in the revised manuscript.

Revision in main text

Despite recent advancements, active learning methodologies remain crucial for perpetually expanding the explorable space in material discovery. This principle aligns with the recent findings by Merchant et al., who demonstrated that expanded dataset enhances model performance in material science. Similarly, our experimental validation of PSCF and PSCFM confirms their systematic identification of superior perovskites. Furthermore, the accuracy of these predictions progressively improves with each iteration. Moving forward, we aim to significantly augment the current dataset,

further enhancing the framework's stability. Additionally, we plan to incorporate models with stronger interpretability to gain deeper insights into the OER process.

Data availability. The data points we have collected are all detailed in the supplementary data section, encompassing the gathered data, sources of the data, information on data post-encoding, and the points of prediction. The first sheet of the supplementary data excel file presents detailed description. To clarify it, we have modified the description in manuscript.

Revision in main text

Due to the limited availability of perovskite oxide data for OER, we additionally collected data for non-OER perovskite oxides (see Supplementary Data for a comprehensive list of data collected and predicted in this work).

2. Can the catalysts with the best predicted activity (especially those with an overpotential of less than 300 mV) be verified experimentally?

Response:

Thanks for your comment. Throughout our OER performance prediction process, only one single data point was predicted to have a minimum predicted value below 300 mV ($\text{Pr}_{0.7}\text{Sr}_{0.3}\text{Fe}_{0.5}\text{Zr}_{0.5}$ in second round with minimum value 286.48 mV and mean value of 333.69 mV). The majority of predicted points with outstanding performance clusters is located in the region with mean value at 320-340 mV. This distribution pattern stems from the fact that merely 3 out of 91 collected points exhibit an overpotential below 300 mV, and 54 of 91 points surpasses 400 mV.

According to previous work, in our data pre-processing step, a standard scaler was applied to enhance predictability, which resulted in the normalization of values, particularly smoothing out those outlier points, i.e., below 300 mV. The standard score z of a sample x is calculated as:

$$z = \frac{x - u}{s}$$

where u is the mean of the training sample and s is the standard deviation. These flattened edged data are caused by an inherent bias within the dataset itself. Given our implementation of an active learning strategy, we anticipated that, with ongoing research and the integration of new exemplary material points in future, more points with overpotential below 300 mV will be discovered. To clarify it, we have modified the description in manuscript.

Revision in Supplementary Notes 2

The data were all preprocessed and normalized using a standard scaler. The standard value z of a sample x is calculated as:

$$z = \frac{x - u}{s} \tag{25}$$

where u is the mean of the training sample and s is the standard deviation.

Revision in main text

Despite recent advancements, active learning methodologies remain crucial for perpetually expanding the explorable space in material discovery. This principle aligns with the recent findings by Merchant et al., who demonstrated that expanded dataset enhances model performance in material science. Similarly, our experimental validation of PSCF and PSCFM confirms their systematic identification of superior perovskites. Furthermore, the accuracy of these predictions progressively improves with each iteration. Moving forward, we aim to significantly augment the current dataset, further enhancing the framework's stability. Additionally, we plan to incorporate models with stronger interpretability to gain deeper insights into the OER process.

3. EPR results should be provided to demonstrate the presence of high oxygen vacancy concentration in PSCF and PSCFM.

Response:

Thanks for your comment. In our manuscript, we have shown XPS results in Figure 4a that both PSCF and PSCFM obtain high oxygen vacancy concentration. Also, as shown in Supplementary Fig. 24f, electrochemical oxygen intercalation in PSCF and PSCFM was examined using CV experiments conducted in an Ar saturated 6 M KOH solution. The observed redox peaks arise from the insertion and extraction of oxygen ions into and from the oxygen-vacant sites.

According to the suggestions from reviewer, we have supplemented the manuscript with extra experiment for the detection and verification of high oxygen vacancy concentrations. The existence of oxygen vacancies in perovskites PSCF and PSCFM was further revealed by electron paramagnetic resonance (EPR) spectra. As shown in Figure R1, the oxygen vacancy signal was detected at $g = 2.003$ obtained by fitting with Bruker Xenon software, indicating the high oxygen vacancy content and well matched with the XPS pattern of O 1s. As described in the article, vacant oxygen sites facilitate nucleophilic attack of OH^- and promote O-O bonding. Previous studies have demonstrated that oxygen vacancies in transition metal oxides induce the formation of new electronic states through hybridization of O 2p and metal 3d orbitals within the bandgap (*Nat. Commun.* **2022**, 13, 1, 2191). These states directly contribute to the enhanced adsorption of intermediates on O-vacancies and the improved electronic conductivity.

Figure R1. EPR spectra of perovskites PSCF and PSCFM.

4. The theoretical activities of Co, Fe and Mn sites in PSCF and PSCFM should be carefully compared.

Response:

Thanks for your comment. We agree with the reviewer that the theoretical activities of Co, Fe and Mn sites in PSCF and PSCFM should be carefully compared. The DFT calculation results of the absorption energy of OH⁻ on Co and Fe cationic sites in PSCF and Co, Fe and Mn cationic sites in PSCFM have been included in the Supporting Information, respectively (Please see *Supplementary Fig. 35*).

The original text is shown as follows:

*As shown in Supplementary Fig. 35, the absorption energy of OH⁻ on Co and Fe cationic sites in PSCF is calculated to be -1.836 eV and -0.471 eV, respectively, indicating that Co sites are the preferable absorption sites. Similarly, Co is also the most energetic favorable absorption site for PSCFM (-2.391 eV on Co, -1.093 eV on Fe and -1.583 eV on Mn). Specifically, in the traditional AEM pathway, the potential determining step (PDS) is the deprotonation of *OOH to form *OO (Intermediate state 3, IS3), which results in a calculated overpotential of 0.71 V on PSCF.*

Based on results shown above, the Co was selected as the reactive site for further simulation.

5. The calculation paths of AEM and LOM seem to be unusual. In common AEM calculations, the initial model is always an unsaturated metal site. In the LOM path, some paths are merged, for example from *OH to Ov-OO, involving both deprotonation and oxygen migration, and the OH-OH to O₂ path is confusing. Please provide the basis for the calculation paths.

Response:

Thanks for your comment. As shown in our manuscript, we performed DEMS on ¹⁸O isotope labelled samples, which provided direct evidence for the involvement of lattice oxygen during the OER process. Therefore, we concluded that the catalyst follows the lattice oxygen oxidation (LOM) mechanism during the reaction. In the meantime, we have revised the potentially misleading parts of the Figure 5 for better clarification of all reaction intermediates in both AEM and LOM. Please see the revised *Figure 5*.

The LOM proceeds through a non-concerted proton-electron transfers step that involves the active site of the metal cation and lattice oxygen. Alexis Grimaud et al. showed by ¹⁸O isotope labelling on-line electrochemical mass spectrometry that part of the oxygen produced during the catalytic OER reaction of perovskite metal oxides is derived from lattice oxygen. The oxidation of this lattice oxygen has different oxygen precipitation activities at different pH, suggesting that the OER reaction involves non-concerted proton-electron transfers (*Nat Chem.* **2017**, 9, 5, 457-465).

LOM differs significantly from conventional AEM. This dynamic change allows for greater efficiency and control in the oxygen evolution process. However, in contrast to conventional AEM, the catalytic surface in the LOM process is no longer thermodynamically stable, but changes dynamically during oxygen evolution. To achieve the OER cycle, it is crucial that the catalyst surface undergoes oxidation, exchange, and release of lattice oxygen ligands. The lattice oxygen oxidation mechanism involves a reaction pathway for the oxygen precipitation reaction under alkaline

conditions. This pathway typically includes oxidant adsorption, where OH^- is adsorbed onto the surface of the catalyst, possibly by adsorption onto oxygen vacancies or metal sites. The adsorbed OH^- then reacts with water molecules to form hydroxide ions and oxygen. This step is the initiating step of the oxygen precipitation reaction. Lattice oxygen plays a crucial role in the oxidation mechanism and provides oxygen atoms to produce oxygen molecules. As the reaction progresses, the oxygen generated is released from the catalyst surface, resulting in the final product of the oxygen evolution reaction. The release of the gaseous O_2 product generates an O_{vac} site, which is refilled by OH^- . Next, the substance undergoes the adsorption of OH^- followed by deprotonation. This occurs through the pathway $\text{O}_{\text{vac}}-\text{OH} \rightarrow \text{OH}-\text{OH} \rightarrow * \text{OH}$. As described in the manuscript, the mechanism still takes the single metal site as the catalytic center to adsorb OH^- and follows the deprotonation step, (i.e., single-metal-site mechanism (SMSM), consistent with the path in this reference, in Figure R2e, *Energy Environ. Sci.* **2021**, 14, 9, 4647-4671). This is the pathway we follow for our DFT simulation.

Figure R2. Different reaction pathways for OER. (a) Schematic illustration of the proposed AEM pathway of OER in alkaline media on an active metal site. The lattice oxygen and oxygen from the electrolyte are marked in black and red colors, respectively. (b) The scaling relation between the binding energies of $*\text{OOH}$ and $*\text{OH}$ on various TMOs. (c) OER volcano plot for various TMOs against the oxygen binding strength ($\Delta G^*\text{O}-\Delta G^*\text{OH}$). (d-f) Schematic illustrations of three alternative pathways of LOM in alkaline media with different catalytic centers. (d) Oxygen-vacancy-site mechanism (OVSM), (e) single-metal-site mechanism (SMSM), and (f) dual-metal-site mechanism (DMSM). The chemically inert lattice oxygen, active lattice oxygen involving OER and oxygen from the electrolyte are marked in black, blue and red colors, respectively, and \square represents lattice O_{vac} . (b and c) are reproduced with permission from ref. 20. Copyright 2016, Nature Publishing Group. (*Energy Environ. Sci.* **2021**, 14, 9, 4647-4671)

Reviewer #2 (Remarks to the Author):

This manuscript reported a novel transfer learning paradigm centered on the cation information to predict undiscovered perovskite oxides for alkaline OER. 13 candidates of representative perovskite oxides were identified and synthesized to form pure perovskite structures. The as-synthesized $\text{Pr}_{0.1}\text{Sr}_{0.9}\text{Co}_{0.5}\text{Fe}_{0.5}\text{O}_3$ (PSCF) and $\text{Pr}_{0.1}\text{Sr}_{0.9}\text{Co}_{0.5}\text{Fe}_{0.3}\text{Mn}_{0.2}\text{O}_3$ (PSCFM) exhibited low OER activity with overpotentials of 327 mV and 315 mV at 10 mA cm⁻², respectively. The authors performed in situ/operando measurements and DFT calculations to confirm the different oxide path mechanisms between the PSCF and PSCFM catalysts. However, lots of conclusions of this work were made without solid experimental supports. Therefore, this manuscript is recommended for publication on Nature Communications after addressing the following issues:

Response:

Thanks for your comment. We have discussed the questions in detail and made replies, especially regarding the structure of the perovskite catalyst involved in this paper. We believe these modifications can improve the manuscript's overall quality.

1. To demonstrate the advantage of four and five metals, the authors need to prepare three and two metals-based catalysts for comparison.

Response:

Thanks for your comment. We experimentally verified more than 30 different ratios from more than 5 million prediction points. The 10 phase-pure perovskites confirmed by XRD results contain one three metals-based catalyst $\text{Pr}_{0.1}\text{Ba}_{0.9}\text{FeO}_3$ (PBF). Please see *Supplementary Fig. 5*, *Supplementary Fig. 6* and *Table S6*. In our data collection, we have included 15 candidates that are in the catalog of three and two metals-based catalysts. The data points we have collected are all detailed in the supplementary data section, encompassing the gathered data and sources of the data. The first sheet of the supplementary data excel file presents detailed description. Also, a new class of transition metal oxide, high-entropy oxide, is receiving booming attention, which renders the potential to manipulate the electronic/structural configuration of solid materials through the control of configurational entropy (*Nat. Catal.* **2021**, 4, 1, 62–70; *Adv. Mater.* **2019**, 31, 26, 1806236). For these reasons, we hope to pay more attention to four and five metals-based compounds catalysts in this work.

Electrochemical LSV evaluation confirmed an overpotential of 399.4 mV for PBF, which is much higher than 327 mV for PSCF. This comparison proves the high catalytic activity of four and five metals-based perovskites (PSCF and PSCFM) predicted in this paper. The other three metals-based perovskites predicted in this work were not successfully synthesized under the same conditions, probably due to the difficulty for phase formation. This part of data was not shown due to space limitations. In addition, the two metals-based perovskites were not predicted in this work.

2. What were the contents of Pr, Sr, Co, Fe, Mn and O elements in the as-synthesized materials? The authors only used XPS to characterize the catalysts. However, XPS is a surface sensitive technique. Therefore, the EDS spectra should be given.

Response:

Thanks for your comment. We acknowledge that it is not rigorous to determine and analyze the element proportion only by XPS, because XPS is a surface-sensitive semi-quantitative analysis technique, which may be more suitable for surface interface chemistry research. In response to your

concerns and to more rigorously analyze the content of each element in the material, we conducted EDS spectroscopy on the samples before and after the OER test. Please refer to the revised Figure R1 and Table R1 below. EDS mappings (Figure R1) confirm the uniform distribution of the elements in the samples without any second-phase segregation. As shown in Table R1, we calculated the average element composition proportion of PSCF and PSCFM based on the spectroscopic results which well matched with the XPS results.

Figure R1. SEM images of (a) PSCF and (c) PSCFM. The EDX mapping results prove the uniform distribution of all elements.

Table R1. Content of each element in PSCF and PSCFM measured by EDS.

	Pr	Sr	Co	Fe	Mn	O
PSCF	2.31 At%	16.38 At%	10.81 At%	10.3 At%	-	60.2 At%
PSCFM	2.36 At%	16.37 At%	10.92 At%	6.53 At%	4.34 At%	59.48 At%

At%: atomic ratio

3. The Faradic efficiency during OER should be provided.

Response:

Thanks for your comment. In respond to the comment, gas chromatography (GC) was used to measure the purity of dilution products produced by the research-grade electrodes for calculating the Faraday efficiency. The experiment protocol could be referred to previous work (*Nat. Commun.* **2023**, 14, 1, 1158). We have revised the experimental procedure in the manuscript to include the part on measuring the Faraday efficiency of the catalysts as follows (*Supplementary Fig. 19 and Table S10*).

Revision in main text

The Faraday efficiency of PSCFM and PSCF anode catalysts was determined by measuring the concentration of gaseous products using gas chromatography based on an efficient four-electron reaction process.. As shown in Supplementary Fig. 19, the high Faraday efficiency over 96% could

be achieved for all measured points, suggesting the good OER selectivity of both electrocatalysts.

Revision in experimental procedure

The electrochemical reaction portion of Faraday efficiency (FE) measurement was performed in a miniature two-compartment electrochemical cell, separated by a piece of Nafion membrane (ACS Catal. 2018, 8, 12, 11342-11351). The oxygen concentration (c) was calculated from the GC peak areas (S) was calculated using Equation as follows:

$$c = 21\% \times \frac{Sp2}{Sp1} \quad (5)$$

where $p1$ was a contrast signal of Oxygen in the air (21%), $p2$ was the oxygen signal produced by the oxygen evolution reaction. Before the reaction, 1M KOH solution was purged with He gas. Next, the electrolytic water reaction was carried out at 0.1 A prior to GC measurement. The generated gas was transported to the anode chamber by He gas at the rate of 20 mL min⁻¹, and then discharged into the gas chromatograph to obtain the peak signal. On the basis of this, the Faraday efficiency can be obtained using the following formula:

$$FE = \frac{4c \times V \times F}{V_m \times I \times t} \quad (6)$$

where V was the volume of the collected gas, V_m was the molar volume constant of the gas, F was the Faraday's constant, I was the application constant current and t was the time.

Revision in supplementary information

Table S10. Faraday efficiency of PSCF and PSCFM.

	1 min	5 min	10 min	30 min	1 h
PSCF	99.89%	96.03%	99.12%	98.25%	99.69%
PSCFM	98.50%	99.78%	97.73%	98.71%	99.05%

Supplementary Fig. 19. The Faraday efficiency of PSCF and PSCFM, tested at different

electrolysis times (1 min, 5 min, 10 min, 30 min and 1 h).

4. The stability test with a current density of 20 mA cm^{-2} and a testing time of 10 h was far not enough. Only using such lower current density is not meaningful. Larger current densities and much longer stability test close to industrial application should be performed to further illustrate the reveal application potential of this catalyst.

Response:

Thanks for your comment. We acknowledge the importance of demonstrating the performance and durability of the reported catalysts in real water electrolyzers (membrane electrode assembly (MEA) mode).

In response to your suggestion, we have conducted additional experiments specifically focusing on the stability of the PSCF and PSCFM catalyst. The experiments involved assessing the performance under practical conditions, specifically at the current density of 30 mA/cm^2 and 50 mA/cm^2 for over 80 hours, please see *Figure 3i–j*. Obviously, the electrolyzer with PSCFM anode and Pt/C cathode has only a slight voltage vibration, implying its good OER stability. Also, we added the corresponding experiment procedure for performing the MEA measurement in revised manuscript.

Revision in main text

The durability of PSCFM was also evaluated in an alkaline water electrolyzer (membrane electrode assembly (MEA) mode) (Figure 3i–j). Clearly, the electrolyzer equipped with a PSCFM anode and a Pt/C cathode exhibited only slight voltage vibration during continuous galvanostatic measurement at 30 mA cm^{-2} and 50 mA cm^{-2} for at least 80 h, implying its promising stability for practical applications.

PSCFM/NF and Pt/C/NF were used as the anode and the cathode, respectively, to prepare a zero-gap alkaline water electrolyzer assembly. Electrochemical measurements were performed with a CHI 760E electrochemical station (Chenhua, Shanghai) at room temperature. The catalyst ink for the electrode was manufactured by dispersing 10 mg of catalyst powder and 1 mL of anhydrous ethanol and 50 μL of Fumion FAA-3-SOLUT-10. In addition, catalyst ink was dropped on a piece of nickel foam surface, yielding catalyst mass loading of 1 mg cm^{-2} and 3 mg cm^{-2} . The two electrodes were separated by an alkaline battery film, which was immersed in a 6 M KOH solution for 12 h prior to use. The scanning rate was 10 mV/s, and the LSV was recorded from 0 to 2.4 V. To assess the long-term stability, CP tests were carried out at a constant current density of 30 mA cm^{-2} and 50 mA cm^{-2} , respectively.

Fig. 3. Experimental validation and performance evaluation of predicted electrocatalysts.

a, The Rietveld refinement of the X-ray powder diffraction (XRD) patterns of PSCF and PSCFM. **b**, Representative TEM, HAADF-STEM image and atomic-scale elemental maps of Pr, Sr, Co, Fe, and Mn in PSCFM. **c**, LSV curves of various electrocatalysts in 1 M KOH electrolyte. **d**, Electrochemical double-layer capacitance (Cdl) plot. **e**, Tafel slope plots. **f**, Response of the charge transfer resistance (Rct) vs applied voltage. **g**, Galvanostatic test of PSCF and PSCFM in three-electrode configuration at 20 mA cm⁻². **h**, Galvanostatic test of PSCF and PSCFM in two-electrode configuration at 10 mA cm⁻². The 20% Pt/C was employed as cathodic electrode. **i**, LSV curve for the electrolyzer in a 6 M KOH solution. **j**, Stability test of the electrolyzer at a current density of 30 mA cm⁻² and 50 mA cm⁻², respectively.

5. Characterizing the oxygen vacancies using O 1s XPS data was not rigorous. The authors may refer to the following references: Surface Science, 2021, 712, 121894. Since Vo plays an important role in regulate the OER activity, can the authors tune the concentration of Vo?

Response:

Thanks for your comment. In response to your concern, we have revised the manuscript for possible

misrepresentation. The valence state and electronic structure of adjacent oxygen atoms are affected due to the deficiency of the oxygen atom. The binding energy of the oxygen vacancy reflects the binding energy of oxygen atoms within the diameter range of several affected neighbouring atoms due to the defect. We have also utilized other characterization tools to exhibit the enrichment of surface oxygen vacancies. In the manuscript, it was described as follows.

To further corroborate the high oxygen vacancy concentration in our materials, electrochemical oxygen intercalation in PSCF and PSCFM was examined using CV experiments conducted in an Ar saturated 6 M KOH solution. The observed redox peaks arise from the insertion and extraction of oxygen ions into and from the oxygen-vacant sites (Supplementary Fig. 24f). The PSCFM exhibits a larger current density in the intercalation regime, indicating its abundance of sites for oxygen intercalation.

To gain further insights into the physical nature of the predicted electrocatalysts for the OER, the oxygen K-edge XAS spectra in TEY mode were measured for PSCF and PSCFM (Fig. 5a). The spectra collected in TEY mode are surface sensitive (~ couple nanometers) due to the limited penetration depth of electrons. The pre-edge peaks below ~530 eV correspond to the oxygen hole states at the conduction band minimum induced by the high valent metal. Peaks A and B reflect the degree of hybridization between the oxygen 2p state and the transition metal 3d state, while peak C represents the interaction between the oxygen 2p state and the Sr 4d state, and peak D corresponds to the mixed state of the transition metal 4sp orbitals. The normalized intensity and energy position of peaks A and B can be used to characterize the covalent degree of the metal-oxygen bonding, which is a crucial factor influencing oxygen adsorption and redox processes. Clearly, the PSCFM exhibits a pre-edge peak of O K-edge at lower energy compared to PSCF, reducing the energy difference between the redox potentials of OH/O₂ and CBM, thereby facilitating electron transfer associated with OER.

Also, we have supplemented the data for the detection and verification of high oxygen vacancy concentrations. The existence of oxygen vacancies in perovskites PSCF and PSCFM was further revealed by electron paramagnetic resonance (EPR) spectra. As shown in Figure R2, the oxygen vacancy signal was detected at $g = 2.003$ obtained by fitting with Bruker Xenon software, indicating the highest oxygen vacancy content and well matched with the XPS pattern of O 1s.

Figure R2. EPR spectra of perovskites PSCF and PSCFM.

The activity of materials is dependent on their electronic structure. Previous studies have shown that catalysts' electronic structure can be regulated through metal ion doping, mechanical strain, or interface regulation. Oxygen vacancies are commonly found in various oxide materials and can significantly impact their structure and performance. Regulating oxygen vacancies can be achieved by controlling the local crystal and electronic structure of the material. Strategies for regulating oxygen vacancies include temperature, atmosphere, atom, ligand or other defect doping, reaction solution concentration, and pressure regulation (*Prog. Chem.* **2023**, 35, 4, 543). For example, Zhuang et al. prepared FeCo_x oxide nanosheets with a high oxygen vacancy concentration (OV-rich) denoted as Fe_xCo_y-ONS (x/y denotes the molar ratio of Fe/Co) to catalyse OER. NaBH₄ was used as a reducing agent. The Fe₁Co₁-ONS mass activity reaches 54.9 A g⁻¹ at an overpotential of 350 mV, with a Tafel slope of 36.8 mV g⁻¹. These values are superior to those of industrial RuO₂, crystalline Fe₁Co₁-ONP, and most reported OER catalysts. The material's thin layer of atoms promotes electron transfer, while the surface O_v enhances electrical conductivity and facilitates H₂O adsorption to nearby Co³⁺ sites (*Adv Mater.* **2017**, 29, 17).

In this work, we can tune the formation of oxygen vacancies by regulating B-site metal cation doping or metal ion vacancies. This may produce more active sites and facilitate the oxygen evolution reaction. However, the manipulation of the electronic structure of a specific material composition is not the major premise of this work which focus on a new transfer learning approach to efficiently screen the undiscovered materials.

Moreover, the formation of oxygen vacancies cannot be fully predicted through theoretical calculations, as there are various scenarios that may arise during experimentation and synthesis. Additionally, oxygen vacancy defects are highly complex and challenging to control. The catalytic performance of the system can only be improved by effectively controlling the content, morphology, structure, and location distribution of oxygen vacancies. This work does not make any predictions about oxygen vacancies. However, oxygen vacancies can alter the local coordination environment of the electrocatalyst by affecting the energy band structure of the material, thereby influencing its electrochemical properties. This can be a promising area for further research.

Revision in main text

Fig. 4a compares the O 1s XPS spectra of different samples to distinguish the different surface oxygen species. Deconvolution reveals four well-fitted peaks: lattice oxygen at 528.1 eV (P1), O⁻ at 529.5 eV (P2), OH/CO₃²⁻ at 531.5 eV (P3), and adsorbed water (H₂O) at 533.1 eV (P4). Notably, PSCFM exhibited a higher percentage of P2 and P3 (79.6%) compared to PSCF (73.6%), suggesting PSCFM has a higher content of oxygen vacancy-related surface absorbed oxygen species. Vacant oxygen sites facilitate nucleophilic attack of OH⁻ and promote O-O bonding. Previous studies have demonstrated that oxygen vacancies in transition metal oxides induce the formation of new electronic states through hybridization of O-2p and metal 3d orbitals within the bandgap. These states directly contribute to the enhanced adsorption of intermediates on O-vacancies and the improved electronic conductivity.

6. The authors focus on the role of Mn and Co elements in oxide path mechanisms. What role do the

other three metals play?

Response:

Thanks for your comment. It is well known that transition metal oxides with perovskite structures typically contain rare and alkaline earth metal elements at the A-site and transition metal elements at the B-site. The majority of A-sites possess ionic properties, and usually do not affect the electronic structure near the Fermi level. Therefore, they are not commonly employed as active sites for catalysis due to their instability. However, the A-site elements possess a large ionic size ($>1 \text{ \AA}$) and are essential components for supporting the angle-shared B-site octahedral framework. The B site is typically occupied by transition metals with favorable thermodynamic stability (such as Mn, Fe, Co, and Ni), which exert a positive influence on OER catalysis. The TMs at the B-site are commonly referred to as "active sites" due to their ability to regulate lattice structure, introduce additional energy levels into the material's band structure to impact electronic configuration, and control defects by means of element doping in the perovskite structure, such as oxygen vacancies and cationic arrangements, which can significantly affect ion transport performance and material stability (*Chem. Rev.* **2015**, 115, 18, 9869-9921; *Energy Environ. Sci.* **2015**, 8,5, 1404-1427).

The DFT calculation results of the absorption energy of OH^- on Co and Fe cationic sites in PSCF and Co, Fe and Mn cationic sites in PSCFM have been included in the Supporting Information, respectively (see *Supplementary Fig. 35*). DFT calculations indicate that the Co site is the preferable absorption site in PSCF. Additionally, Co is the most energetically favorable absorption site in PSCFM, with a value of -2.391 eV , compared to Fe's -1.093 eV and Mn's -1.583 eV .

7. The authors should evaluate the performance of the as-synthesized PSCFM catalyst in anion exchange membrane (AEM) electrolyzer.

Response:

Thanks for your comment. We acknowledge the importance of demonstrating the performance and durability of the reported catalysts in real water electrolyzers (membrane electrode assembly (MEA) mode).

In response to your suggestion, we have conducted additional experiments specifically focusing on the stability of the PSCF and PSCFM catalyst. The experiments involved assessing the performance under practical conditions, specifically at the current density of 30 mA/cm^2 and 50 mA/cm^2 for 80 h, please see *Figure 3i-j*. Obviously, the electrolyzer with PSCFM anode and Pt/C cathode has only a slight voltage vibration, implying its good OER stability. Also, we added the corresponding experiment procedure for performing the MEA measurement in revised manuscript.

Revision in main text

The durability of PSCFM was also evaluated in an alkaline water electrolyzer (membrane electrode assembly (MEA) mode) (Figure 3i-j). Clearly, the electrolyzer equipped with a PSCFM anode and a Pt/C cathode exhibited only slight voltage vibration during continuous galvanostatic measurement at 30 mA cm^{-2} and 50 mA cm^{-2} for at least 80 h, implying its promising stability for practical applications.

PSCFM/NF and Pt/C/NF were used as the anode and the cathode, respectively, to prepare a zero-gap alkaline water electrolyzer assembly. Electrochemical measurements were performed with a CHI

760E electrochemical station (Chenhua, Shanghai) at room temperature. The catalyst ink for the electrode was manufactured by dispersing 10 mg of catalyst powder and 1 mL of anhydrous ethanol and 50 μL of Fumion FAA-3-SOLUT-10. In addition, catalyst ink was dropped on a piece of nickel foam surface, yielding catalyst mass loading of 1 mg cm^{-2} and 3 mg cm^{-2} . The two electrodes were separated by an alkaline battery film, which was immersed in a 6 M KOH solution for 12 h prior to use. The scanning rate was 10 mV/s , and the LSV was recorded from 0 to 2.4 V. To assess the long-term stability, CP tests were carried out at a constant current density of 30 mA cm^{-2} and 50 mA cm^{-2} , respectively.

Fig. 3. Experimental validation and performance evaluation of predicted electrocatalysts. *a*, The Rietveld refinement of the X-ray powder diffraction (XRD) patterns of PSCF and PSCFM. *b*, Representative TEM, HAADF-STEM image and atomic-scale elemental maps of Pr, Sr, Co, Fe, and Mn in PSCFM. *c*, LSV curves of various electrocatalysts in 1 M KOH electrolyte. *d*, Electrochemical double-layer capacitance (Cdl) plot. *e*, Tafel slope plots. *f*, Response of the charge transfer resistance (Rct) vs applied voltage. *g*, Galvanostatic test of PSCF and PSCFM in three-electrode configuration at 20 mA cm^{-2} . *h*, Galvanostatic test of PSCF and PSCFM in two-electrode configuration at 10 mA cm^{-2} . The 20% Pt/C was employed as cathodic electrode. *i*, LSV curve for

the electrolyzer in a 6 M KOH solution. j, Stability test of the electrolyzer at a current density of 30 mA cm⁻² and 50 mA cm⁻², respectively.

Reviewer #3 (Remarks to the Author):

This work reported a transfer learning paradigm to predict perovskite oxides for oxygen evolution reaction. The authors found that $\text{Pr}_{0.1}\text{Sr}_{0.9}\text{Co}_{0.5}\text{Fe}_{0.5}\text{O}_3$ (PSCF) and $\text{Pr}_{0.1}\text{Sr}_{0.9}\text{Co}_{0.5}\text{Fe}_{0.3}\text{Mn}_{0.2}\text{O}_3$ (PSCFM) exhibited good OER activity. They attributed this to the incorporation of Mn into PSCF strengthened the stability of Co reactive sites and lowered the reaction barrier. Electrochemical measurements revealed the coexistence of the adsorbent evolution mechanism and lattice oxygen mechanism for O-O coupling. The main drawback of this study is that the structure of catalysts is not well understood. In addition, several important issues need to be addressed:

Response:

Thanks for your comment. We have addressed all the following questions point-by-point in detail. We believe these modifications could promote overall quality of the manuscript.

1. The authors stated a transfer learning paradigm. What is the major difference between it and density functional theory calculations and machine learning? What are the key parameters that determine the prediction results?

Response:

Thanks for your comment. To leverage existing knowledge and accelerate the discovery of optimal OER materials, we employed transfer learning, a powerful machine learning approach that builds upon pre-existing knowledge to gain new insights (*Journal of Big data*, **2016**, 3, 1-40). This methodology hinges on identifying and exploiting the inherent similarities between known and unknown knowledge domains. In our research, we implemented transfer learning through two key strategies: data augmentation and a global ensemble technique. Data augmentation involved expanding the OER material dataset by incorporating non-OER materials, leveraging the well-established similarities within perovskite materials. This enriched dataset provided a stronger foundation for the model. The global ensemble strategy further enhanced the learning process by utilizing data similarity to modulate the weight assigned to each model's prediction. This approach fine-tuned the overall prediction based on the relevance and relationships between data points.

In comparison to Density Functional Theory (DFT) calculations, this methodology offers the distinct advantage of facilitating a cost-effective and high-throughput screening. Via two rounds of screening, we have successfully predicted over 20 million data points—an undertaking that presents substantial challenges for traditional DFT. Given the valuable insights of reaction mechanisms afforded by DFT calculations, our future endeavors will integrate DFT-derived information to enrich the dataset, thereby enhancing the comprehensiveness of our analysis. We have claimed the high computational cost in the main text in introduction section of the manuscript:

To accelerate the exploration of novel and efficient perovskite oxide-based OER catalysts, high-throughput density functional theory (DFT) calculations have emerged as a promising alternative. Nevertheless, these calculations often require prior knowledge of specific algorithms or methods, hindering data unification across different systems and limiting the universal applicability of the results under various conditions. Additionally, while computational costs are lower than experimental work, they are not negligible, further complicating seamless plug-and-play implementation.

For linear regression and its more intricate counterparts, like symbolic regression (e.g., SISSO (*Physical Review Materials*, **2018**, 2(8), 083802)), that provide explicit equations, it's feasible to ascertain definitive feature importance. However, these interpretable models, often referred to as 'glass box' models, are limited in their ability to accurately capture complex relationships. On the other hand, complex 'black box' models, while potentially more powerful for intricate systems, often suffer from the challenge of interpreting feature importance measures extracted from these models, which can hinder interpretability and the discovery of new materials – a key objective in our research (*Nat. Catal.* **2022**, 5(3), 175-184). Therefore, to prioritize the identification of novel materials, we opted to leverage the wealth of feature information available. During the cation encoding phase, we generated an extensive array of features and later employed an auto-encoder with shortcut connections (AESC), aiming to preserve as much feature information as feasible. This allows us to divert attention away from feature importance towards maximizing the model's predictive capacity for material discovery. As we claimed in the introduction part of the manuscript:

Some efforts have focused on using ML to identify highly relevant descriptors, thereby simplifying the system and accelerating the prediction process. However, the ML algorithms based on feature selection and simplification often eliminate less-significant descriptors, inevitably leading to information loss and diminished prediction accuracy. Moreover, when analyzing the relative importance of the same dataset, different algorithms frequently produce inconsistent results. Consequently, the reliability of feature importance interpretations in the absence of domain expertise is often questionable.

Concerning model interpretability, we aim to explore models with enhanced interpretability in our future work to uncover deeper insights. To clarify it, we have modified the description in manuscript.

Revision in main text

Despite recent advancements, active learning methodologies remain crucial for perpetually expanding the explorable space in material discovery. This principle aligns with the recent findings of Merchant et al., who demonstrated that expanded dataset enhances model performance in material science. Similarly, our experimental validation of PSCF and PSCFM confirms their systematic identification of superior perovskites. Furthermore, the accuracy of these predictions progressively improves with each iteration. Moving forward, we aim to significantly augment the current dataset, further enhancing the framework's stability. Additionally, we plan to incorporate models with stronger interpretability to gain deeper insights into the OER process.

2. The authors stated that the enhanced OH- absorption and surface oxygen vacancies played a critical role in promoting O-O coupling during the OER process. However, more evidence for the strength of OH- absorption and concentration of surface oxygen vacancies should be provided. Some key spectra characterizations are missing.

Response:

Thanks for your comment. We have supplemented the data for the detection and verification of high oxygen vacancy concentrations. The existence of oxygen vacancies in perovskites PSCF and PSCFM was further revealed by electron paramagnetic resonance (EPR) spectra. As shown in Figure R1, the

oxygen vacancy signal was detected at $g = 2.003$ obtained by fitting with Bruker Xenon software, indicating the highest oxygen vacancy content and well matched with the XPS pattern of O 1s. As described in the article, vacant oxygen sites facilitate nucleophilic attack of OH^- and promote O-O bonding. Previous studies have demonstrated that oxygen vacancies in transition metal oxides induce the formation of new electronic states through hybridization of O-2p and metal 3d orbitals within the bandgap. These states directly contribute to the enhanced adsorption of intermediates on O-vacancies and the improved electronic conductivity.

Figure R1. EPR spectra of perovskites PSCF and PSCFM.

To address the issue of OH^- adsorption, we evaluated the adsorption capacity of OER intermediates using methanol as a probe, because an active OER electrochemical material should possess an optimal affinity for OH^- intermediates during OER. Our results confirm the hypothesis of enhanced reactant adsorption. We compared the catalysts' performance in both the methanol oxidation reaction and the precipitated oxygen reaction, as detailed in the manuscript.

To corroborate the hypothesis of enhanced reactant adsorption, methanol was used as a probe to assess the adsorption capacity of OER intermediates. As OH^- is an electrophilic OER intermediate, it readily reacts with nucleophilic methanol. Consequently, the increase in current density between the methanol oxidation reaction (MOR) and OER polarization curves correlates with the surface coverage of OH^- . Prior to analysis, the C_{dl} values for both PSCF and PSCFM catalysts were determined in MOR (Supplementary Fig. 23). The C_{dl} for PSCF (2.15 mF cm^{-2}) and PSCFM (2.66 mF cm^{-2}) in MOR are similar to those achieved in OER. This indicates that the influence of ECSA on the current increase in MOR is negligible. As shown in Supplementary Fig. 23 and Fig. 4b, the significantly higher MOR current density of PSCFM (2.5 times that of PSCF) clearly demonstrates a stronger affinity for OH^- and thus higher OH^- coverage on the PSCFM surface. These results agree with the in situ EIS analysis (Fig. 3d), which suggest enhanced OH^- adsorption on the catalyst.

Upon reviewer's request, we further evaluate the adsorption of reactants on the Co sites of PSCF and the Mn sites of PSCFM during OER, both materials were examined on the basis of the Laviron analysis. The detailed results could be found as follows. Briefly, PSCFM shows a larger redox constant ($K_s = 0.17 \text{ s}^{-1}$) than that of PSCF ($K_s = 0.16 \text{ s}^{-1}$), suggesting the stronger coupling strength of *OH intermediates to the surface.

Revision in main text

Furthermore, to investigate the absorption of reactants on the PSCF and PSCFM during OER, Laviron analysis was conducted for both materials. As shown in Supplementary Fig. 25, the steady-state redox currents associated with OH⁻ transfer show a linear correlation with the square root of potential scan rates in the CV (1 to 35 mV s⁻¹) curves. Notably, PSCFM shows a larger redox constant ($K_s = 0.17 \text{ s}^{-1}$) compared to PSCF ($K_s = 0.16 \text{ s}^{-1}$), suggesting a stronger binding strength between *OH intermediates and the surface (*J. ELECTROANAL. CHEM.* **1979**, 100, 1-2, 263-270; *Adv. Funct. Mater.* **2021**, 31, 25, 2100614).

Revision in main text

The redox constant (K_s) of catalysts was calculated using Laviron Equation as follows:

$$E_c = E_{1/2} - (RT/\alpha nF) \times \{\ln(\alpha nF/RTK_s) + \ln(v)\} \quad (7)$$

where E_c is the reduction potential of metal redox, $E_{1/2}$ is the formal potential of metal redox, R is the universal gas constant, T is the temperature in kelvin, n is the number of electrons transferred, α is the transfer coefficient, K_s is the rate constant of metal redox, and v is the scan rate in the CV measurements (*Sci Adv.* **2018**, 4, 3, eaap7970).

Revision in Supplementary Information

Supplementary Fig. 25. K_s of PSCF and PSCFM obtained from Laviron analysis. Plots of the redox peak currents densities versus the square root of scan rates of (a) PSCFM and (c) PSCF. Laviron analyses of (b) PSCFM and (d) PSCF.

3. The authors demonstrated that the incorporation of Mn into PSCF to form Co-Obridge-Mn motifs strengthened the stability of CO reactive sites and lowered the reaction barrier. It is worth noting that the chemical valence of Co is 2.46 and 2.49 for PSCFM and PSCF based on the XPS results. How to understand the interaction between Co and Mn?

Response:

Thanks for your comment. The interaction between Co and Mn can be described as a synergistic effect, which alters the catalytic performance of the site and lowers the reaction barrier of Co. DFT calculations reveal that the interfacial structure (Co-O-Mn) with enhanced mobility of lattice oxygen acts as the intrinsic active center to facilitate the activated adsorption (the rate-determining step). Doping Mn modifies the density distribution of electronic states near the Co-O bridge bond, thereby influencing the adsorption capacity of Co at this position. Overall, by regulating the electronic structure, the interaction between Co and Mn impacts the characteristics of Co reaction sites in perovskite structures through localization effects and generating synergistic effects to enhance both reaction activity and stability. In our manuscript, we also demonstrate charge transfer characteristics within the system using Bader charge calculations, illustrating that formation of a Co-O_{bridge}-Mn motifs leads to electron redistribution and co-localization around Co. Supported by DFT calculations and XPS tests, we provide evidence for how the synergistic effect of Co and Mn promotes oxygen evolution reaction as follows:

As shown in Fig. 5e, the Bader charges of Co and Mn are positive, indicating their roles as electron donors. The higher electron density on the oxygen bridge O_{bridge} (from 1.03- to 1.08-) is also evident. Notably, the substitution of Co with Mn to form Co-O_{bridge}-Mn leads to electron redistribution and electron colocalization around Co, accompanied by a decrease in the Co Bader charge from 0.7+ to 0.63+. This finding aligns with the XPS results. The reduced Bader charge reveal an orderly decrease in the degree of ionization, which can significantly enhance resistance to cation leaching.

Consistent with these experimental findings, the DFT calculations (Fig. 5f) also confirm that the absorption of *OH is thermodynamically more favorable on V_o in the Co-V_o-Mn motif (-2.63 eV) than in the Co-V_o-Co counterpart (-2.46 eV), facilitating the refilling of O_{lattice} and generation of BO₆. Additionally, the Co element exhibits a significant valence increase from 2.49 to 2.59 in PSCF, likely due to substantial surface dissolution and reconstruction (Supplementary Table 9). In contrast, the valence of Co in PSCFM only show pretty subtle variation (0.01), indicating a more dynamically stable site.

4. Based on the characterization and DFT calculations, it was suggested that Co was the active site, and the introduction of Mn improved the stability of Co centers. However, Fe is also important for the adsorption and activation of oxygen species. Is there any evidence to exclude this? Confirmation of active center needs additional evidence.

Response:

Thanks for your comment. In response to your concern, we wish to discuss the importance of Fe at

first. As we all know, Fe plays a critical role in enhancing the alkaline oxygen evolution reaction (OER) activity of catalysts based on transition metals. Researchers have extensively studied the influence of iron on the catalytic activity of transition metal-based catalysts, and it has been observed that iron can effectively enhance the electrochemical properties and stability of these catalysts during the OER.

Prof. Shannon W. Boettcher and colleagues found that controlling the concentration of Fe ions in the electrolyte helped maintain a relatively constant amount of Fe species on the anode catalyst surface. The interaction between the Ni-based catalyst and Fe significantly reduces the electrolyte voltage; however, the stability of the Ni-based catalyst is compromised during repeated start-stop cycles in the electrolysis system. In contrast, although exhibiting slightly inferior catalytic performance, the Co-based catalyst demonstrates superior stability. They studied the incorporation and OER-activation by foreign electrolyte ions into electrodeposited NiO_xH_y and CoO_xH_y films and discovered that under controlled oxidative conditions the incorporation can be limited to surface sites. In the case of Fe on NiO_xH_y , we used this approach to demonstrate both intrinsically high OER activity (at optimal surface-Fe loading), and a new fundamental picture emphasizing cooperative effects between multiple Fe sites that share oxidative charge. The computations show that new low overpotential pathways for OER are possible through synergistic interaction between multiple Fe species and host-metal atoms whereby oxidative charge can be favorably delocalized and stabilized (*Nat. Commun.* **2023**, 14, 1, 7688).

Kowalski et al. employed state-of-the-art electronic structure calculations and thermodynamic modeling to investigate the phenomenon that the electrocatalytic activity of NiOOH is increased by about three orders of magnitude by the addition of 25% Fe (close to the solubility limit of Fe in NiOOH) to NiOOH. It was shown that at low concentrations, Fe exists in a low spin state. Only this spin state explains the large solubility limit of Fe as well as the similarity of Fe-O and Ni-O bond lengths measured in the Fe-doped NiOOH phase. The low spin state makes the surface Fe sites highly active for OER (*Nat. Commun.* **2023**, 14, 1, 3498).

Based on the characterization and DFT calculations, it was suggested that Co was the active site, and the introduction of Mn improved the stability of Co centers as shown in the manuscript. The DFT calculation results of the absorption energy of OH^- on Co and Fe cationic sites in PSCF and Co, Fe and Mn cationic sites in PSCFM have been included in the Supporting Information, respectively (see *Supplementary Fig. 35*).

The original text is as follows:

As shown in Supplementary Fig. 35, the absorption energy of OH^- on Co and Fe cationic sites in PSCF is calculated to be -1.836 eV and -0.471 eV, respectively, indicating that Co sites are the preferable absorption sites. Similarly, Co is also the most energetic favorable absorption site for PSCFM (-2.391 eV on Co, -1.093 eV on Fe and -1.583 eV on Mn). Specifically, in the traditional AEM pathway, the potential determining step (PDS) is the deprotonation of $^\text{OOH}$ to form $^*\text{OO}$ (Intermediate state 3, IS3), which results in a calculated overpotential of 0.71 V on PSCF.*

5. The authors suggested that the catalysts showed good electrolysis durability. While the structure of catalysts after the stability test was missing. More evidence for the stability of PSCF and PSCFM

should be provided and analyzed the samples after the reaction.

Response:

Thanks for your comment. In response to your comment, we have added a characterization of the structure of the catalysts after OER to support the good stability of the catalysts. We conducted tests on PSCF and PSCFM in the constant flow mode of 20 mA/cm² for 10 hours, and obtained the catalyst samples after the reaction. The scanning electron microscopy (SEM, Figure R2 and Figure R3) images manifest that the catalyst samples before and after OER have similar morphology. After the 10 h long-term OER, the structure is almost unchanged further demonstrating the electrochemical stability. As you can see in the manuscript, the catalyst's morphology and crystal structure can remain intact even after the stability test.

While XRD and SEM data revealed that the overall crystal structures and morphologies of PSCF and PSCFM remained intact after the OER stability test (Supplementary Fig. 38), the HR-TEM image of PSCFM after OER measurement showed an amorphous surface region with a depth of ~2 nm, adjacent to the bulk with a highly ordered atomic arrangement (Fig. 5c).

Also, to gain more information for the active oxides on the surface after the long-term OER, we collected the EDS mappings on the oxides (Figure R2b,d and Figure R3b,d). EDS analysis further reveal a uniform distribution of Pr, Sr, Co, Fe and Mn elements. The proportions of the elements have not changed significantly.

Figure R2. SEM images and corresponding EDS elemental mappings on the sample region of (a,b) PSCF and (c,d) Post-OER PSCF.

Figure R3. SEM images and corresponding EDS elemental mappings on the sample region of (a,b) PSCFM and (c,d) Post-OER PSCFM.

Table R1. Content of each element in PSCF and PSCFM measured by EDS.

	Pr	Sr	Co	Fe	Mn
PSCF	2.31 At%	16.38 At%	10.81 At%	10.3 At%	-
Post-OER PSCF	2.20 At%	14.38 At%	10.00 At%	9.98 At%	-
PSCFM	2.36 At%	16.37 At%	10.92 At%	6.53 At%	4.34 At%
Post-OER PSCFM	1.46 At%	11.11 At%	7.76 At%	4.44 At%	3.03 At%

At%: atomic ratio

REVIEWERS' COMMENTS

Reviewer #1 (Remarks to the Author):

The revised version is excellent and i would like to suggest the acceptance.

Reviewer #2 (Remarks to the Author):

The authors have fully addressed the comments and this manuscript is recommended for publication now.

Reviewer #2 (Remarks on code availability):

Yes

Reviewer #3 (Remarks to the Author):

This is an updated version based on my revision request. I can see the authors have properly addressed the concerns by conducting additional research and discussion. Therefore i am happy it to be accepted now.